EMBO
Molecular Medicine

# Murine Langerin[+] dermal dendritic cells prime CD8[+] T cells while Langerhans cells induce cross-tolerance

Vincent Flacher[1,2,†], Christoph H Tripp[1,2], David G Mairhofer[1], Ralph M Steinman[3], Patrizia Stoitzner[1,a,**], Juliana Idoyaga[3,‡] & Nikolaus Romani[1,2,a,*]

## Abstract

Skin dendritic cells (DCs) control the immunogenicity of cutaneously administered vaccines. Antigens targeted to DCs via the C-type lectin Langerin/CD207 are cross-presented to CD8[+] T cells *in vivo*. We investigated the relative roles of Langerhans cells (LCs) and Langerin[+] dermal DCs (dDCs) in different vaccination settings. Poly(I:C) and anti-CD40 agonist antibody promoted cytotoxic responses upon intradermal immunization with ovalbumin (OVA)-coupled anti-Langerin antibodies (Langerin/OVA). This correlated with CD70 upregulation in Langerin[+] dDCs, but not LCs. In chimeric mice where Langerin targeting was restricted to dDCs, CD8[+] T-cell memory was enhanced. Conversely, providing Langerin/OVA exclusively to LCs failed to prime cytotoxicity, despite initial antigen cross-presentation to CD8[+] T cells. Langerin/OVA combined with imiquimod could not prime CD8[+] T cells and resulted in poor cytotoxicity in subsequent responses. This tolerance induction required targeting and maturation of LCs. Altogether, Langerin[+] dDCs prime long-lasting cytotoxic responses, while cross-presentation by LCs negatively influences CD8[+] T-cell priming. Moreover, this highlights that DCs exposed to TLR agonists can still induce tolerance and supports the existence of qualitatively different DC maturation programs.

**Keywords** CD8[+] T-cell responses; dendritic cells; Langerhans cells; skin; tolerance

**Subject Categories** Immunology

## Introduction

Dendritic cells (DCs) are professional antigen-presenting cells, specialized to take up protein antigens and process them into peptides that they load onto their MHC molecules. Once activated, DCs migrate to lymph nodes and interact with antigen-specific T cells there. Thus, DCs are essential to initiate adaptive immune responses (Palucka *et al*, 2010; Steinman, 2012). Over the last decade, the role of the different DC subsets has been intensively addressed in a variety of immune responses. Functional specializations have been identified, although their generalization and translation from mouse to humans (Cohn *et al*, 2013) is complicated by the phenotypic plasticity of DCs under the influence of their environment. In particular, identifying which DCs can efficiently prime cytotoxic responses remains a controversial question that needs to be resolved, because such immune responses are essential to fight against viral infections and tumors. Indeed, increasing evidence from clinical studies emphasizes the role of CD8[+] T cells in the control of tumors and the prolongation of patient survival (Galon *et al*, 2006; Mellman *et al*, 2011; Dhodapkar *et al*, 2014).

Presentation of exogenous antigens to CD8[+] T cells (cross-presentation) is probably not exclusive to one DC subset (Dickgreber *et al*, 2009; Segura *et al*, 2013). In the mouse, a consistent body of evidence indicates a dominating contribution of lymph-node-resident CD8[+] DCs in cross-presentation (Pooley *et al*, 2001; Joffre *et al*, 2012). However, skin DCs such as epidermal Langerhans cells (LCs) (Stoitzner *et al*, 2006; Flacher *et al*, 2010) and Langerin[+] dermal DCs (dDCs) (Bedoui *et al*, 2009; Henri *et al*, 2010) can also cross-present. *In vivo*, the relative involvement of skin DCs versus CD8[+] DCs in priming efficient immune responses does not depend only on the targeted subset, but also on other factors, including the type, amount and formulation of antigen, the site of immunization,

1  Department of Dermatology and Venereology, Innsbruck Medical University, Innsbruck, Austria
2  Oncotyrol Center for Personalized Cancer Medicine, Innsbruck, Austria
3  Laboratory of Cellular Physiology and Immunology and Chris Browne Center for Immunology and Immune Diseases, The Rockefeller University, New York, NY, USA
   *Corresponding author. Tel: +43 512 504 28559; Fax: +43 512 504 67 28559; E-mail: nikolaus.romani@i-med.ac.at
   **Corresponding author. Tel: +43 512 504 23016; Fax: +43 512 504 67 28592; E-mail: patrizia.stoitzner@i-med.ac.at
   [a]These authors contributed equally to this work.
   R.M.S. passed away on September 30, 2011.
   [†]Present address: Laboratory of Immunopathology and Therapeutic Chemistry/Laboratory of Excellence MEDALIS, CNRS UPR3572, Institut de Biologie Moléculaire et Cellulaire, Strasbourg, France
   [‡]Present address: Department of Microbiology and Immunology, Stanford University School of Medicine, Stanford, CA, USA

the endocytic route and intracellular degradation pathways, and the maturation signals received by DCs (Delamarre *et al*, 2003; Spörri & Reis e Sousa, 2005; Blander & Medzhitov, 2006; Tacken *et al*, 2007; Idoyaga *et al*, 2011; Chatterjee *et al*, 2012).

When efficiently primed, CD8$^+$ T cells upregulate the IL-7 receptor/CD127 and establish themselves as CD44$^{high}$ CD62L$^+$ central memory CD8$^+$ T cells in the lymphoid organs (Belz & Kallies, 2010). Upon secondary exposure to the antigen, memory CD8$^+$ T cells rapidly differentiate into IFN-γ-producing effector cells with the ability to reach the periphery. On the other hand, self-specific CD8$^+$ T cells must be kept under control, a phenomenon known as peripheral cross-tolerance (Gill & Tan, 2010). Indeed, cross-presentation does not always result in priming of a cytotoxic response, since initial proliferation of CD8$^+$ T cells can be followed by deletion of the T-cell clone, instead of differentiation into effector T cells (Bonifaz *et al*, 2002). The deletion of self-specific CD8$^+$ T-cell clones is instrumental to avoid deleterious cytotoxic responses (Kurts *et al*, 1997). Induction of cross-tolerance can also result from anergy or suppression by regulatory CD4$^+$ T cells (Lutz & Kurts, 2009; Gill & Tan, 2010). The underlying mechanisms are not fully understood to date and probably depend on the co-stimulatory signals delivered by DCs at the time of antigen presentation.

Antibodies (Ab) recognizing C-type lectins (Sancho & Reis e Sousa, 2012) such as DEC-205 and Langerin have been used to enhance internalization of antigens by specific DC subsets, resulting in potent CD4$^+$ and CD8$^+$ T-cell responses (Bonifaz *et al*, 2002; Idoyaga *et al*, 2008, 2013). Recently, this strategy has also been applied to initiate antigen-specific tolerance, which occurred when antigen was presented to T cells by DCs in the steady state (Hawiger *et al*, 2001; Yamazaki *et al*, 2008). Nevertheless, inefficient T-cell priming despite apparent upregulation of molecules considered DC maturation markers (e.g., CD86) has been documented (Lutz & Schuler, 2002; Spörri & Reis e Sousa, 2005; Jiang *et al*, 2007; Platt *et al*, 2010). This implies that 'true' DC maturation (i.e., resulting in immunogenicity) remains to be accurately defined on a molecular level (Joffre *et al*, 2009; Steinman, 2012).

Our previous work identified LCs and Langerin$^+$ dermal DCs (dDCs) as prominent transporters of intradermally injected antibodies recognizing C-type lectins (Flacher *et al*, 2010, 2012). Antigen targeting to DCs via Langerin results in activation of antigen-specific transgenic CD8$^+$ T cells *in vivo* (Idoyaga *et al*, 2008, 2009), but this work was performed in [C57BL/6 × BALB/c] F1 mice where Langerin is expressed not only by LCs and dDCs, but also by CD8$^+$ DCs in skin-draining lymph nodes (Flacher *et al*, 2008). Therefore, the outstanding ability of CD8$^+$ DCs in cross-presentation may mask the contribution of cutaneous Langerin$^+$ DC subsets to CD8$^+$ T-cell immunity. Conversely, LCs targeted *in vivo* with ovalbumin (OVA)-coupled anti-Langerin Ab (Langerin/OVA) did not stimulate the proliferation of transgenic CD4 and CD8 T cells *in vitro* (Flacher *et al*, 2010). These contrasting results led us to further investigate T-cell responses depending on Langerin-mediated antigen capture by skin DCs in a refined experimental model *in vivo*.

We show here that targeting skin DCs through Langerin has variable outcomes. In C57BL/6 mice that have low if any Langerin expression in CD8$^+$ DCs, cutaneous immunization with Langerin/OVA conjugates triggered either efficient cytotoxic immune responses or hyporesponsiveness of antigen-specific CD8$^+$ T cells. Strikingly, cross-tolerance developed only in the presence of the

adjuvant imiquimod, but not under steady state conditions. When we used chimeric mice to target antigen via Langerin exclusively into either LCs or dermal DCs, we observed that LCs induced cross-tolerance while Langerin$^+$ dDCs stimulated memory and cytotoxic responses of CD8$^+$ T cells. Our results introduce novel insights into the different roles of Langerin$^+$ skin DCs in the development of CD8$^+$ T-cell responses.

# Results

## Langerin targeting requires strong adjuvants to allow development of cytotoxic responses and CD8$^+$ T-cell memory

Previously, we had compared the capacity of LCs to present antigen internalized through different C-type lectins. After DEC-205-targeted mature LCs emigrated out of epidermal explants, they induced proliferation of both OVA-specific CD4$^+$ and CD8$^+$ transgenic T cells *in vitro*. On the other hand, LCs targeted through Langerin failed to present antigen to T cells (Flacher *et al*, 2010).

To extend these observations *in vivo*, we monitored endogenous killing responses against target cells loaded with the OVA MHC class I peptide SIINFEKL. Targeting antigen to lectin receptors requires additional adjuvants to generate efficient antigen-specific T-cell responses (Hawiger *et al*, 2001; Bonifaz *et al*, 2002). Intradermal injection of OVA-coupled anti-DEC-205 monoclonal Ab (DEC/OVA) yields high rates of target cell killing when a cream containing TLR7 ligand imiquimod was topically applied to the immunization site (Flacher *et al*, 2012). However, Langerin targeting by OVA-coupled anti-Langerin monoclonal Ab (clone L31; Langerin/OVA) in similar conditions failed to trigger endogenous cytotoxic responses (Fig 1A and Supplementary Fig S1A). Moreover, vaccination with imiquimod and Langerin/OVA did not protect mice from the growth of OVA-expressing transplanted B16 melanoma, thereby severely impairing survival of animals bearing the tumor (Supplementary Fig S1B and C).

A combination of the TLR3 ligand poly(I:C) with an agonist anti-CD40 Ab (pIC/40) has been successfully used to generate CD8$^+$ T-cell immunity after DEC-205 and Langerin targeting (Bonifaz *et al*, 2004; Sancho *et al*, 2008; Trumpfheller *et al*, 2008; Idoyaga *et al*, 2011). We injected this adjuvant intradermally together with Langerin/OVA and observed a potent cytotoxic response, in contrast to imiquimod (Fig 1A). To better understand this difference, we transferred CFSE-labeled OVA-specific CD8$^+$ T cells from CD45.1$^+$ OT-I transgenic mice, 1 day before immunization. Similar to our previous observations (Idoyaga *et al*, 2008), proliferation of OT-I CD8$^+$ T cells was observed 6 days after i.d. injection of Langerin/OVA, regardless of the adjuvant (Fig 1B). This is due to the considerable sensitivity of OT-I CD8$^+$ T cells to minute amounts of antigen (Choi *et al*, 2009). However, OT-I CD8$^+$ T cells showing a high number of divisions (more than 6) were in largest proportions when the adjuvant was pIC/40. Furthermore, the receptor for IL-7 (IL-7R/CD127), which is required for survival of memory T-cell precursors (Belz & Kallies, 2010), was only upregulated when the mice had been treated with pIC/40 (Fig 1C and D).

Because the initial proliferation burst may not lead to establishment of a memory response, we looked for surviving OT-I CD8$^+$ T

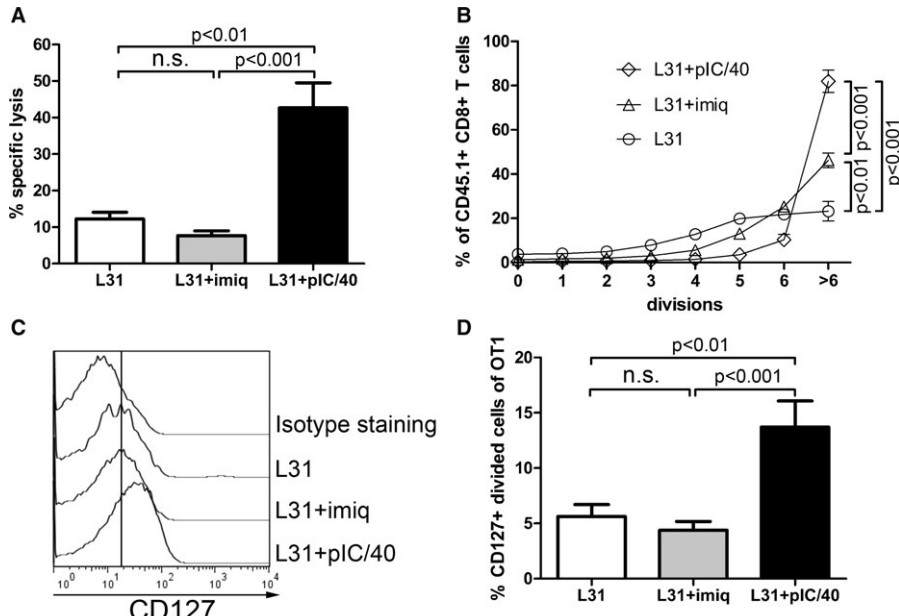

**Figure 1.  Langerin targeting with poly(I:C) and anti-CD40 efficiently triggers CD8$^+$ T-cell responses.**

A    *In vivo* killing of OVA-loaded target cells. C57BL/6 mice were immunized i.d. into both ears with 0.5 µg Langerin/OVA (L31) alone or in addition to imiquimod (+imiq) or poly(I:C) and anti-CD40 (+pIC/40). Seven days later, CFSE-labeled OVA-loaded target cells and CTO-labeled unloaded control cells were transferred i.v. Specific lysis of OVA-loaded target cells was measured in the blood 2 days later. Values from individually analyzed mice are pooled from two independent experiments (L31: three mice; L31+imiq: 10 mice; L31+pIC/40: six mice) and compared using one-way ANOVA (*P* < 0.0001) followed by Tukey's test (n.s.: non-significant, *P* > 0.05).

B–D   Proliferation and differentiation of OVA-specific transgenic CD8$^+$ T cells. CD8$^+$ T cells purified from [OT-I × Ly5.1] F1 mice were labeled with CFSE and injected i.v. into C57BL/6 mice. The next day, mice were immunized i.d. into both ears with 0.5 µg Langerin/OVA (L31) alone or in addition to imiquimod (+imiq) or poly(I:C) and anti-CD40 (+pIC/40). Six days later, skin-draining lymph nodes were digested, and CD45.1$^+$ CD8$^+$ T cells were analyzed by flow cytometry for proliferation and expression of IL-7R/CD127. Values from individually analyzed mice are pooled from three independent experiments (L31: six mice; L31+imiq: nine mice; L31+pIC/40: five mice) and compared using one-way ANOVA followed by Tukey's test (n.s.: non-significant, *P* > 0.05). (B) Proportions of cells that underwent 0–6 or more cycles of division (ANOVA: *P* < 0.0001). (C) Representative histogram plots of CD127 stainings. The vertical line depicts the geometric mean intensity of fluorescence when immunizing with Langerin/OVA alone. (D) Proportion of CD127$^+$ divided cells (ANOVA: *P* = 0.0004).

cells in skin-draining lymph nodes 8 weeks after immunization. At this late time point, only rare OT-I T cells, including CD44$^{low}$ naive cells, could be found in mice treated with Langerin/OVA alone or together with imiquimod (Fig 2A). However, when Langerin/OVA immunization was performed in the presence of pIC/40 treatment, abundant OT-I CD8$^+$ T cells were still present, with > 90% displaying the CD44$^+$ CD62L$^+$ phenotype of central memory T cells (T$_{CM}$). Consistent with this, OT-I T cells from mice treated with Langerin/OVA plus pIC/40 had a dramatically increased potential for reactivation. *In vitro* restimulation of lymph node cells with the OVA MHC I peptide SIINFEKL resulted in differentiation of T$_{CM}$ cells into CD62L$^{low}$ effector T cells with considerably stronger synthesis of IFN-γ as compared to untreated or imiquimod-treated mice (Fig 2B and C).

**Treatment with different adjuvants does not alter distribution of anti-Langerin targeting antibodies**

Upon injection into the skin, the anti-Langerin L31 clone binds to Langerin$^+$ dermal DCs, LCs (Idoyaga *et al*, 2008) and, in mice with BALB/c genetic background, to CD8$^+$ DCs (Idoyaga *et al*, 2009). Inflammatory stimuli may affect antigen distribution to DC subsets, for instance by a possible *de novo* Langerin expression in potently cross-presenting lymph node-resident CD8$^+$ DCs of C57BL/6 mice.

To address this, we injected a fluorescent full-length anti-Langerin L31 antibody or isotype control in the same amount and route as OVA-coupled conjugates (Supplementary Fig S2). CCR7$^{neg}$ CD8$^+$ lymph node-resident DCs represented less than 0.5% of targeted DCs in any given condition, emphasizing that the vast majority of targeted cells in the lymph nodes comes from the skin. In mice not treated with adjuvant, most of the CD11c$^+$ DCs targeted by fluorescent anti-Langerin antibodies were CCR7$^+$ CD8$^{neg}$ skin-derived DCs (Mean ± SD: day 2, 91.1% ± 8.3; day 4, 83.6% ± 12.1). The distribution of targeting antibody was similar between the different DC subsets regardless of the adjuvant used. No significant difference was observed in mice treated with imiquimod (day 2, 91.7% ± 5.2; day 4, 85.3% ± 4.7) or poly(I:C)/aCD40 (day 2, 91.7% ± 3.1; day 4, 90.2% ± 4.0). Among these targeted skin DCs, we could identify LCs, Langerin$^+$ dDCs, and Langerin$^{neg}$ CD103$^{neg}$ dDCs. However, a fraction of the latter population also captured the isotype control antibody. This clearly suggests a non-specific, Fc Receptor (FcR)-dependent binding of full-length antibodies. Of note, FcR-mediated uptake cannot occur with OVA-coupled conjugates, because they contain a mutation in their FcR-binding site (Clynes *et al*, 2000; Hawiger *et al*, 2001). Altogether, the contribution of CD8$^+$ DCs and Langerin$^{neg}$ dDCs in immune responses triggered by Langerin/OVA conjugates is most likely marginal and independent of the adjuvant used.

                                      

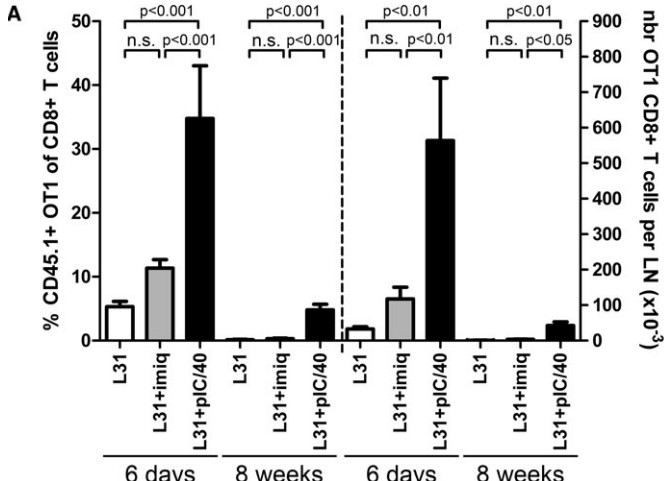

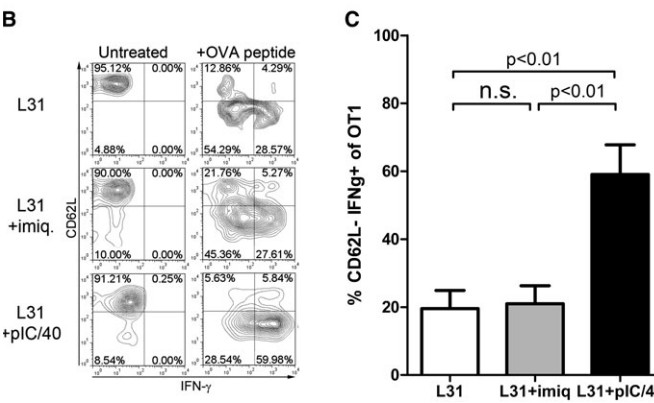

**Figure 2.  Poly(I:C) and anti-CD40 Ab allow generation of memory CD8⁺ T cells after Langerin targeting.**

CD8⁺ T cells purified from [OT-I × Ly5.1] F1 mice were labeled with CFSE and injected i.v. into C57BL/6 mice. The next day, mice were immunized i.d. into both ears with 0.5 μg Langerin/OVA (L31) alone or in addition to imiquimod (+imiq) or poly(I:C) and anti-CD40 (+pIC/40). Data from individually analyzed mice are pooled from three independent experiments and compared using one-way ANOVA followed by Tukey's test (n.s.: non-significant, $P > 0.05$).

A   Six days or 8 weeks after immunization, the proportions (L31: six mice; L31+imiq: nine mice; L31+pIC/40: five mice; ANOVA: $P = 0.0002$ at day 6, $P = 0.0001$ at week 8) and absolute numbers (L31: four mice; L31+imiq: five mice; L31+pIC/40: five mice; ANOVA: $P = 0.0011$ at day 6, $P = 0.0061$ at week 8) of CD45.1⁺ CD8⁺ T cells in skin-draining lymph nodes were evaluated.

B   After 8 weeks, total lymph node cells were exposed overnight to the OVA peptide SIINFEKL. CD62L expression and IFN-γ production were visualized in CD45.1⁺ CD8⁺ T cells by flow cytometry. Representative stainings.

C   Percentage of CD62L-low IFN-γ-producing among OT-I CD8⁺ T cells (L31: four mice; L31+imiq: five mice; L31+pIC/40: five mice; ANOVA: $P = 0.0024$).

## Langerin⁺ dermal DC, but not LCs, upregulate CD70 after treatment with poly(I:C) and anti-CD40

Induction of CD8⁺ T-cell memory depends on factors expressed by DCs at the time of antigen presentation to naive cells, including expression of CD70 (Sanchez et al, 2007) and release of IL-12 (Trinchieri, 2003). Fitting with previous observations (Soares et al, 2007), intradermally injected adjuvant pIC/40 was considerably more potent at inducing expression of CD70 than application of

imiquimod (Fig 3A and B). Interestingly, only Langerin⁺ dDCs, but not LCs, could upregulate CD70 on their surface after pIC/40 treatment. Conversely, IL-12p40 synthesis by either subset of Langerin⁺ DCs was not affected by any adjuvant (Fig 3C and D).

We have also analyzed expression of different factors involved in CD8⁺ T-cell tolerance, namely CD273/PD-L2/B7-DC, CD274/PD-L1/B7-H1, and CD275/ICOSL/B7-H2 (Chen, 2004). We observed somewhat increased expression of CD274/PD-L1 on LCs and Langerin⁺ dDCs in response to both adjuvants, whereas CD273/PD-L2 and CD275/ICOSL remained unchanged (Supplementary Fig S3A and B). Additional factors known to induce T helper 2 responses (CD134/OX40L) (Liu, 2007) or immunoregulation (IL-10) (Boonstra et al, 2006) were not at all expressed in Langerin⁺ DCs/LCs (Supplementary Fig S3C).

## Selective Langerin expression by skin DC subsets in bone marrow chimeric mice

Upon lethal irradiation and reconstitution, only epidermal LCs survive (Merad et al, 2002), while other DCs, including Langerin⁺ dermal DCs, are newly generated from transferred bone marrow (Ginhoux et al, 2007; Poulin et al, 2007) (Supplementary Fig S4A). We took advantage of this and designed bone marrow transfer protocols using Langerin⁻/⁻ (LKO) mice (Kissenpfennig et al, 2005a) to obtain Langerin expression in selected skin DCs, thereby making Langerin targeting only possible in either Langerin⁺ dermal DCs or LCs. Three months after bone marrow transfer, Langerin staining of epidermal sheets (Supplementary Fig S4B) and epidermal cell suspensions (Supplementary Fig S4C) confirmed that LCs remained mostly of recipient origin, although a small proportion appeared to be derived from the bone marrow graft, in accordance with recent results (Nagao et al, 2012). In skin-draining lymph nodes, Langerin⁺ CD103⁺ EpCAMneg dDCs (Henri et al, 2010) were only observed in mice that received wild-type bone marrow grafts (Supplementary Fig S4D). When Langerin⁺ LCs could be observed in epidermal cell suspensions, a corresponding Langerin⁺ CD103neg population was found in skin-draining lymph nodes. In summary, we obtained mice with Langerinneg dDCs and Langerin⁺ LCs (LKO→wt) or, conversely, Langerin⁺ dDCs and Langerinneg LCs (wt→LKO).

## Langerhans cells are not involved in cross-priming of CD8⁺ T-cell responses

Using these chimeric mice, we investigated which Langerin⁺ DC subset is responsible for CD8⁺ T-cell responses after Langerin targeting and robust stimulation by pIC/40. Killing of OVA peptide-loaded target cells was only slightly decreased in mice where only Langerin⁺ dermal DCs could be targeted. In contrast, targeted LCs alone were unable to prime endogenous cytotoxic responses (Fig 4A). This was not due to a lack of antigen cross-presentation, since initial proliferation of transferred OT-I CD8⁺ T cells occurred similarly in LKO→wt and wt→LKO mice (Fig 4B). CD127 upregulation by proliferating cells was also visible in both chimeras (Fig 4C). Finally, 3 weeks after Langerin targeting with pIC/40, CD8⁺ T-cell memory was estimated by IFN-γ synthesis by restimulation with OVA MHC I peptide of OT-I T cells recovered from lymph nodes (Fig 4D). The absence of LC targeting (wt→LKO chimera) led to markedly superior IFN-γ production as compared to chimeric mice

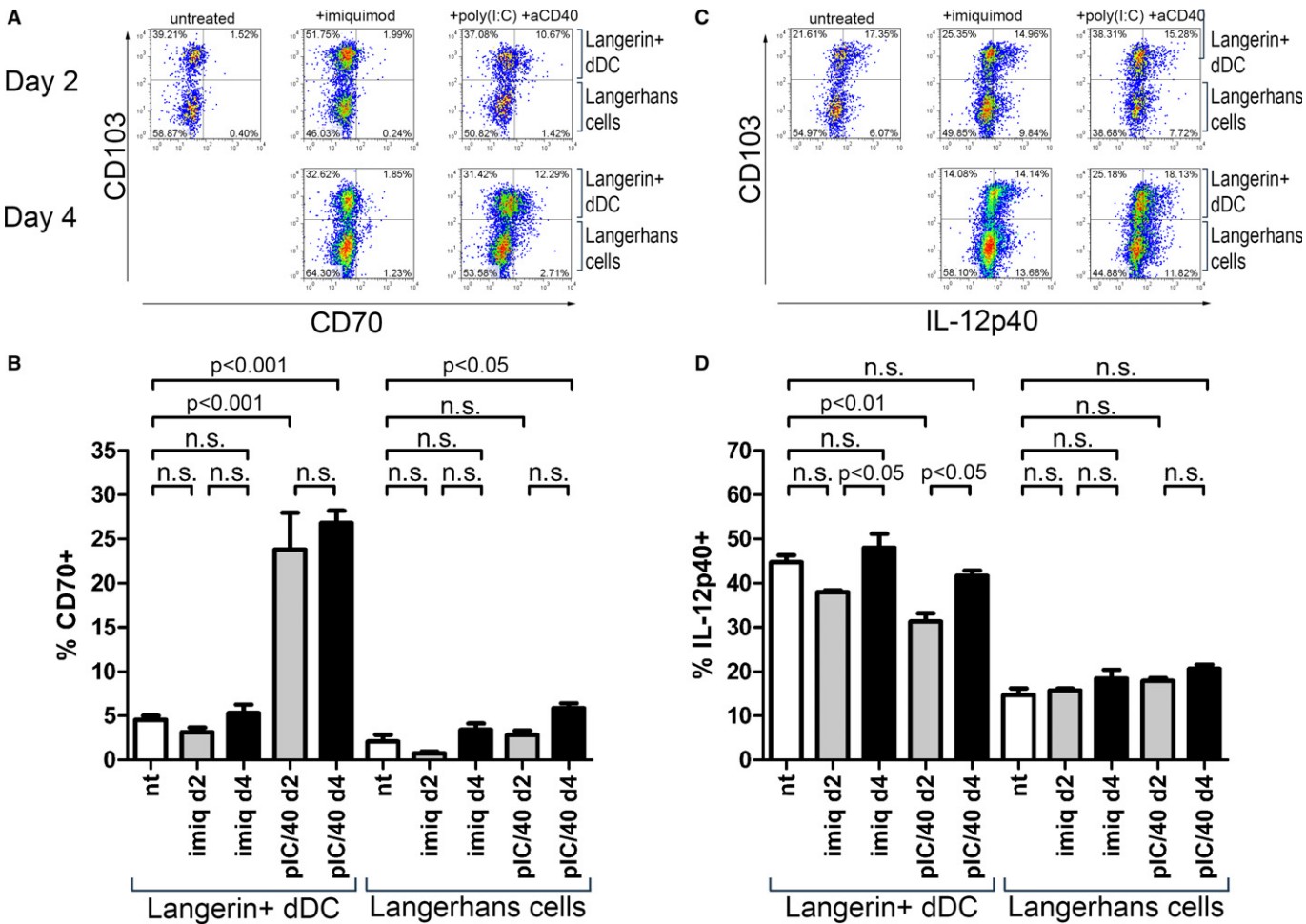

**Figure 3.  Langerin⁺ dDCs, but not LCs, express CD70 upon pIC/40 stimulation *in vivo*.**

Both ears of C57BL/6 mice were treated with imiquimod (imiq), poly(I:C) and anti-CD40 (pIC/40), or left untreated (nt). Two or 4 days later, a cell suspension was obtained from four auricular lymph nodes collected from two identically treated mice, and expression of CD70 and IL-12p40 was evaluated by flow cytometry. Measurements, each comprising cells from two mice, are pooled from two independent experiments (nt: *n* = 7; imiq—day 2: *n* = 4; imiq—day 4: *n* = 4; pIC/40—day 2: *n* = 4; pIC/40—day 4: *n* = 3) and compared using one-way ANOVA followed by Tukey's test (n.s., non-significant, *P* > 0.05).

A    Representative stainings showing surface expression of CD70 in CD11c⁺ Langerin⁺ DCs.

B    Percentages of Langerin⁺ dDCs and LCs expressing CD70 (ANOVA: *P* < 0.0001 and *P* = 0.0003, respectively).

C    Representative stainings showing intracellular expression of IL-12p40 in CD11c⁺ Langerin⁺ DCs after a 3-h incubation with Brefeldin A.

D    Percentages of Langerin⁺ dDCs and LCs expressing IL-12p40 (ANOVA: *P* = 0.0008 and *P* = 0.0995, respectively).

that lacked Langerin on dDCs (LKO→wt), reinforcing the possibility of a stronger priming by Langerin⁺ dDCs. We observed here an unexpectedly low response of wild-type mice. Although IFN-γ production may not perfectly correlate with cytotoxicity, this contrasted with the results of killing assays (Fig 4A). However, it should be noted that the proportion of IFN-γ-producing cells of wild-type mice were typically lower here than in other experiments (i.e., restimulation 8 weeks after immunization, Fig 2C). Altogether, this does not change the overall conclusions of our study and the validity of the side-by-side comparison of the chimeric mice.

**Langerin targeting promotes hyporesponsiveness of CD8⁺ T cells**

To extend our initial findings (Fig 1), we sought to determine if Langerin targeting in the presence of imiquimod could influence the development of subsequent immune responses. We set up a

pre-treatment consisting of intradermal injection of ISO/OVA or Langerin/OVA into one ear, in the presence of imiquimod. One week later, we elicited cytotoxic responses by DEC/OVA and topical imiquimod in the contralateral ear (Supplementary Fig S1; Flacher *et al*, 2012).

Although no primary cytotoxic responses were observed after immunization with imiquimod and either ISO/OVA or Langerin/OVA, pre-treating the mice with the latter had a visible impact on secondary responses. Langerin/OVA pre-treatment led to a 61% decrease of OVA-specific target cell lysis in the blood, as compared with ISO/OVA (Fig 5A). This experiment was repeated in Langerin⁻/⁻ mice (Fig 5B), or in the absence of imiquimod (Fig 5C). In both cases, conditioning with Langerin/OVA did not impair the secondary immune response.

Intradermal immunization with DEC/OVA plus imiquimod in only one ear resulted in relatively low killing rates. Intraperitoneal

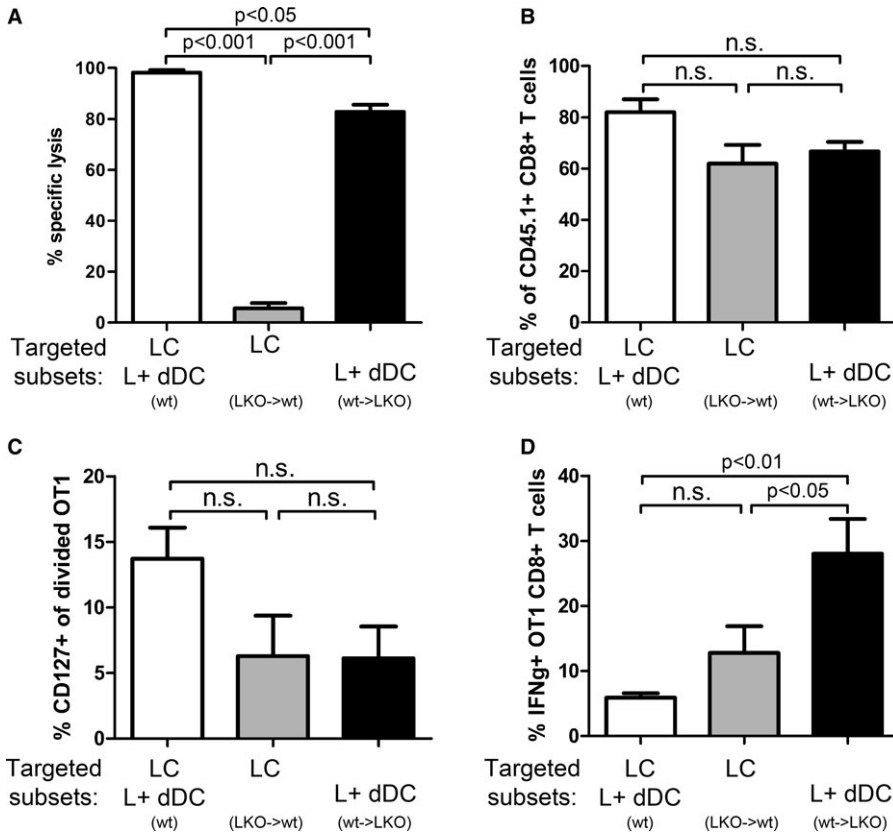

**Figure 4. Langerin⁺ dermal DCs have a dominant role in activation of CD8⁺ T cells.**

A   LKO→wt, wt→LKO chimeric mice, or C57BL/6 wt mice were immunized into both ears with 0.5 µg Langerin/OVA in the presence of poly(I:C) and anti-CD40 Ab. Seven days later, CFSE-labeled OVA-loaded target cells and CTO-labeled unloaded control cells were transferred i.v. Specific lysis of OVA-loaded target cells by the endogenous cytotoxic T cells was measured in lymph nodes 2 days later. Data from individually analyzed mice are pooled from at least two independent experiments (wt: 3 mice; LKO→wt: seven mice; wt→LKO: 16 mice) and compared using one-way ANOVA ($n$ = 3–16; $P$ < 0.0001) followed by Tukey's test (n.s.: non-significant, $P$ > 0.05).

B–D Proliferation and differentiation of OVA-specific transgenic CD8⁺ T cells. CD8⁺ T cells obtained from [OT-I × Ly5.1] F1 mice were labeled with CFSE and transferred i.v. into LKO→B6 or B6→LKO chimeric mice, or wt mice. The following day, both ears were immunized with Langerin/OVA in the presence of poly(I:C) and anti-CD40. Data are pooled from four independent experiments and compared using one-way ANOVA followed by Tukey's test (n.s.: non-significant, $P$ > 0.05). (B,C) Six days later, skin-draining lymph nodes were digested, and CD45.1⁺ CD8⁺ T cells were analyzed by flow cytometry for proliferation (wt: five mice; LKO→wt: five mice; wt→LKO: five mice). Proportions of cells with more than six cycles of division are depicted in (B) ($n$ = 5; ANOVA: $P$ = 0.0598) and proportions of CD127⁺ divided cells in (C) (ANOVA: $P$ = 0.1085). (D) Three weeks after transfer, total skin-draining lymph nodes cells were stimulated overnight with OVA peptide SIINFEKL. The percentage of CD45.1⁺ CD8⁺ T cells producing IFN-γ was evaluated by flow cytometry (individually analyzed mice: wt: 9; LKO→wt: 12; wt→LKO: 12; ANOVA: $P$ = 0.0037).

injection of DEC/OVA and poly(I:C), however, allowed to reach approximately 80% of specific target cell lysis. Despite these high levels, we still observed a potent inhibition following Langerin/OVA conditioning (Fig 5D). Interestingly, when the primary immunization was done under similar conditions, but with pIC/40 instead of imiquimod, targeting with Langerin/OVA did not result in tolerance (Supplementary Fig S5A).

Finally, we extended our observations by following the growth of transplanted B16 tumors expressing OVA. When the conditioning step was performed with ISO/OVA and imiquimod, subsequent intraperitoneal DEC/OVA and poly(I:C) considerably slowed down the growth of implanted B16-OVA tumors (Fig 5E), thereby increasing the lifespan of tumor-bearing animals (Fig 5F). On the other hand, conditioning with Langerin/OVA and imiquimod significantly impaired the therapeutic effect of DEC/OVA tumor treatment,

further confirming induction of antigen-specific hyporesponsiveness.

### Cross-tolerance upon Langerin targeting relies on Langerhans cells, but not Langerin⁺ dermal DCs

To exclude the possibility that an ongoing primary response interferes with the development of secondary cytotoxic responses, we waited for 6 weeks after the pre-treatment to trigger a cytotoxic response using i.p. DEC/OVA and poly(I:C). A potent reduction of the secondary response was still observed in Langerin/OVA-pre-treated mice (Fig 6A, left). In parallel, the proportion of OVA-specific endogenous CD8⁺ T cells, measured by OVA-Kb pentamer staining, was lower in Langerin/OVA-pre-treated mice (Fig 6A, right).

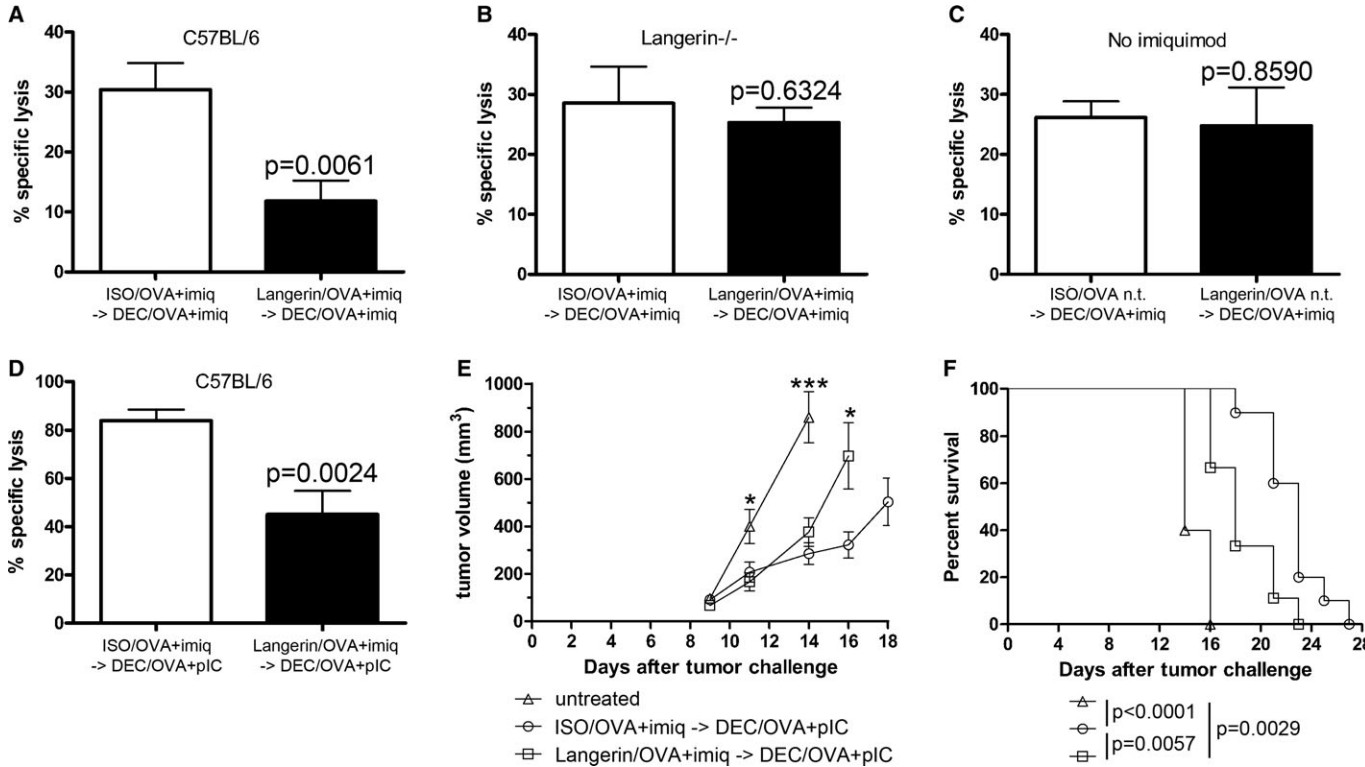

**Figure 5.  Combination of Langerin/OVA and imiquimod decreases endogenous killing responses and impairs anti-tumoral responses against tumor cells expressing ovalbumin.**

A–C  (A) One ear was immunized with Langerin/OVA or isotype control ISO/OVA and treated with imiquimod (imiq). One week later, the contralateral ear was immunized with DEC/OVA and imiquimod. One week later, CFSE-labeled OVA-loaded target cells and CTO-labeled unloaded control cells were transferred i.v. Specific lysis of OVA-loaded target cells was measured in the blood 2 days later (*n* = 7). Data from individually analyzed mice are pooled from at least two independent experiments (ISO/OVA+imiq → DEC/OVA+pIC: seven mice; Langerin/OVA+imiq → DEC/OVA+pIC: seven mice) and compared using Student's unpaired *t*-test. The same experiment was performed (B) in Langerin[−/−] mice (ISO/OVA+imiq → DEC/OVA+pIC: six mice; Langerin/OVA+imiq → DEC/OVA+pIC: six mice) or (C) in C57BL/6 without topical imiquimod application (nt) (ISO/OVA nt → DEC/OVA+pIC: five mice; Langerin/OVA nt → DEC/OVA+pIC: six mice).

D  Both ears were immunized with Langerin/OVA or isotype control ISO/OVA and treated with imiquimod (imiq). One week later, mice were injected i.p. with 2 µg anti-DEC-205/OVA and 50 µg poly(I:C) (DEC/OVA+pIC). One week later, CFSE-labeled OVA-loaded target cells and CTO-labeled unloaded control cells were transferred i.v. Specific lysis of OVA-loaded target cells was measured in the blood 2 days later. Values from individually analyzed mice are pooled from three independent experiments (ISO/OVA+imiq → DEC/OVA+pIC: nine mice; Langerin/OVA+imiq → DEC/OVA+pIC: nine mice) and compared using Student's unpaired *t*-test (n.s.: non-significant, *P* > 0.05).

E, F  Tumor protection assay. Both ears of C57BL/6 mice were immunized with Langerin/OVA or isotype control ISO/OVA and treated with imiquimod, or left untreated. On the same day, $10^5$ B16 melanoma cells expressing OVA were implanted subcutaneously. One week later, mice were injected i.p. with 2 µg anti-DEC-205/OVA and 50 µg poly(I:C) (DEC/OVA+pIC). Tumor growth (E) and survival of recipient mice (F) were monitored three times a week. Data from individually analyzed mice are pooled from two independent experiments (untreated: five mice; ISO/OVA+imiq → DEC/OVA+pIC: 10 mice; Langerin/OVA+imiq → DEC/OVA+pIC: 10 mice). Tumor growth was compared using one-way ANOVA (day 11: *P* = 0.0131; day 14: *P* < 0.0001) followed by Tukey's test (*P* < 0.05; ***P* < 0.001), or using Student's unpaired *t*-test (day 16: *P* = 0.0226). Survival curves were compared using a Mantel–Cox test.

This suggests that imiquimod disturbs the establishment of antigen-specific memory upon uptake of the antigen via Langerin. We performed similar experiments in chimeric mice. A clear inhibition of killing as well as a reduced percentage of OVA-specific endogenous CD8[+] T cells were observed in mice which retained Langerin expression only on epidermal LCs (Fig 6B). Strikingly, when OVA was targeted exclusively to Langerin[+] dDCs, killing rates as well as the proportion of OVA-Kb pentamer[+] CD8[+] T cells remained similar to the control pre-treatment.

Finally, we used our chimera system to investigate whether the tolerogenic influence of LCs was retained with a stronger adjuvant. Thus, we performed the primary immunization with Langerin/OVA in the presence of pIC/40 instead of imiquimod. Similar to wild-type C57BL/6 mice (Supplementary Fig S5A), tolerance induction was absent in control chimera mice, as well as in mice with dDC-restricted Langerin expression (Supplementary Fig S5B). Unexpectedly, when only LCs could be targeted by Langerin/OVA, secondary responses were repressed, suggesting that LCs can exert a tolerogenic function in the presence of either adjuvant. However, in wild-type mice and control chimeras, the tolerogenic contribution of LCs may be obscured or outweighed by the potent priming capacity of Langerin[+] dDCs.

# Discussion

Different types of antigen-presenting cells reside in the skin. For this reason, the potential of the skin as a vaccination site has attracted

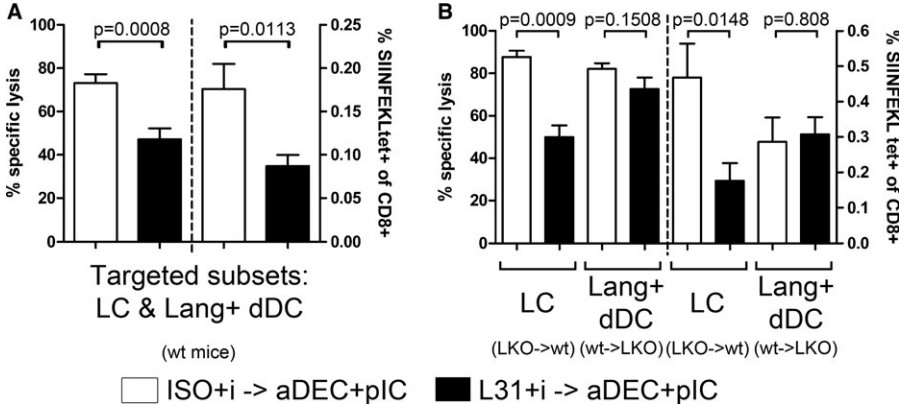

**Figure 6. Impairment of secondary CD8[+] T-cell responses is dependent on targeting of Langerhans cells.**
Both ears of C57BL/6 or chimeric mice were immunized with 0.5 μg Langerin/OVA (L31) or isotype control ISO/OVA and treated with imiquimod (imiq). Six weeks later, mice were injected i.p. with 2 μg anti-DEC-205/OVA and 50 μg poly(I:C) (DEC/OVA+pIC). After 6 days, CFSE-labeled OVA-loaded target cells and CTO-labeled unloaded control cells were obtained from Ly5.1 mice and transferred i.v. Specific lysis of OVA-loaded CD45.1[+] target cells was measured in the blood 1 day later, while endogenous OVA-specific CD8[+] T cells of skin-draining lymph nodes were characterized as CD45.1[−] CD19[−] CD4[−] NK1.1[−] OVA-Kb pentamer[+] CD8[+].

A  C57BL/6 mice: values from individually analyzed mice are pooled from three independent experiments and compared using Student's unpaired *t*-test (ISO+i→aDEC+pIC: 10 mice; L31+i→aDEC+pIC: 10 mice).

B  Chimeric mice: data from individually analyzed mice are pooled from two independent experiments and compared using Student's unpaired *t*-test (LKO→wt: [ISO+i→aDEC+pIC]: four mice, [L31+i→aDEC+pIC]: seven mice; wt→LKO [ISO+i→aDEC+pIC]: 9 mice; [L31+i→aDEC+pIC]: 10 mice).

considerable interest in the last few years. Skin DC subsets have been the subject of many investigations regarding their relative immunological roles. Despite this, their precise contribution to CD8[+] T-cell immunity is not definitely settled. We further examined their capacity of cross-presentation and, importantly, priming of cytotoxic responses in a murine system where antigen uptake is mediated by the Langerin/CD207 receptor present on two important cutaneous DC subsets, namely LCs and Langerin[+] dDCs. We describe here that immunization strategies targeting Langerin generate either cytotoxic responses and long-lived memory or antigen-specific cross-tolerance. Strikingly, the latter did not occur in the steady state, but required application of an adjuvant. Moreover, we found that Langerin[+] dDCs are potent at cross-priming, while LCs are more prone to induce CD8[+] T-cell tolerance.

Differences in the target receptor result in different internalization pathways (Mahnke *et al*, 2000; Cohn *et al*, 2013). In addition, these pathways may be different from one DC subset to another. As a consequence, critical variations in antigen processing and presentation can be expected from different targeting strategies. We have previously found intradermal DEC-205 targeting of OVA to result in potent priming of endogenous CD8[+] T-cell responses in the presence of imiquimod. We have identified Langerin[neg] dDCs (Flacher *et al*, 2012) and CD8[+] DCs (Idoyaga *et al*, 2011) as essential subsets following cutaneous DEC-205 targeting, depending on the mouse strains. These two potent immunostimulatory populations of DCs may represent the key to explain the differences with Langerin targeting in that they might outcompete the tolerogenic activity of the simultaneously targeted DEC-205[+] LCs.

There is evidence for *in vitro* cross-presentation of keratinocyte-derived or exogenously added OVA by LCs (Stoitzner *et al*, 2006; Holcmann *et al*, 2009; Flacher *et al*, 2010), although *in vivo* OT-I proliferation assays rather identified Langerin[+] dDCs as cross-presenting cells (Bursch *et al*, 2009; Henri *et al*, 2010; Igyarto *et al*, 2011). To study the contribution of skin DCs in immune responses,

we and others have previously used diphtheria toxin (DT)-induced depletion of Langerin[+] DCs in Langerin-DTR mice (Kissenpfennig *et al*, 2005b; Poulin *et al*, 2007; Noordegraaf *et al*, 2010; Flacher *et al*, 2012). Unfortunately, this model does not permit reliable depletion of either LCs or Langerin[+] dDCs over several weeks. The time needed for reconstitution of the depleted Langerin[+] DCs is variable and influenced by inflammation in the skin (Ginhoux *et al*, 2006; Nagao *et al*, 2012), possibly leading to misinterpretations when comparing different adjuvants. In addition, 15 days after depletion, LCs are still absent, but absolute numbers of Langerin[+] dDCs remain low as compared to untreated mice (Noordegraaf *et al*, 2010). For these reasons, we have chosen a bone marrow chimeric model (Merad *et al*, 2002; Bursch *et al*, 2009; Shklovskaya *et al*, 2011) that allowed us to selectively target either Langerin[+] dDCs or LCs with Langerin/OVA conjugates.

The absence of endogenous cytotoxic responses observed during exclusive targeting of LCs proves that Langerin[+] dDCs are far superior in priming OVA-specific immunity. Despite this, both chimeras show similar early proliferation of OT-I CD8[+] T cells. The lack of LC promotes the generation of highly sensitive memory CD8[+] T cells (Fig 4D). Furthermore, the induction of tolerance by Langerin/OVA pre-treatment relies solely on LCs (Fig 6B and Supplementary Fig S5B). Taken together, these results confirm that LCs do cross-present antigens to CD8[+] T cells *in vivo*, but are not sufficient for priming cytotoxic responses. Instead, cross-presentation of Langerin-targeted antigen by LCs leads to deletion of antigen-specific T cells. Such properties fit with previous explanations for the tolerogenic role of LCs (Igyarto *et al*, 2011).

A widely accepted paradigm states that immature DCs induce tolerance because they support differentiation of regulatory T cells (Treg) from naïve CD4[+] T cells (Yamazaki *et al*, 2006), while mature DCs promote immunity (Hawiger *et al*, 2001). Imiquimod is an approved drug for the treatment of basal cell carcinoma and genital warts. This TLR7 agonist promotes emigration and maturation of

skin DCs, including LCs (Suzuki *et al*, 2000; Flacher *et al*, 2008). Consequently, imiquimod can trigger potent cytotoxic immune responses (Rechtsteiner *et al*, 2005; Flacher *et al*, 2012) and a psoriasis-like skin inflammation that relies on dDCs (Van der Fits *et al*, 2009; Wohn *et al*, 2013). Despite this, in wild-type mice, imiquimod appears to have a tolerogenic influence, thereby supporting alternatives to the above-mentioned paradigm (Joffre *et al*, 2009). Our observations are direct evidence that the decision between immunity versus tolerance must not be simply equated with DCs being phenotypically mature versus immature, respectively. Intrinsic properties of DC subsets or different maturation programs may lead to tolerogenic DCs which do express known maturation markers, but are unable to prime sustained proliferation of T cells and their differentiation into memory cells (Jiang *et al*, 2007; Platt *et al*, 2010). Here, we extend these findings to another experimental model and specifically to skin DCs.

In murine experimental setups, tolerance induced by LCs has been explained by mechanisms such as induction of Treg (Gomez de Agüero *et al*, 2012), immunosuppressive IL-10 release (Igyarto *et al*, 2009), or incomplete maturation (Azukizawa *et al*, 2011; Shklovskaya *et al*, 2011). We could not detect production of IL-10 in emigrant LCs, and targeting via Langerin did not yield more antigen-specific Treg than targeting via DEC-205 (Idoyaga *et al*, 2013). In line with our observations, when OVA expression was induced in keratinocytes, imiquimod-triggered inflammation was not sufficient to prime OT-I-dependent autoimmunity, and this tolerance to OVA was independent of Treg (Holcmann *et al*, 2009).

Surprisingly, chimeric mice with LC-restricted Langerin expression displayed a strong tolerogenic capacity when treated with either imiquimod or pIC/40. We conclude that, in these chimera at least, Langerin-mediated antigen uptake by LCs alone and subsequent presentation to CD8[+] T cells in skin-draining lymph nodes may be sufficient to promote deletional cross-tolerance regardless of the adjuvant. Nevertheless, in wild-type mice that would correspond to the physiological situation, pIC/40 simultaneously unfolds the strong immunogenic potential of Langerin[+] dDCs, thereby hiding the tolerogenic contribution of LCs, while imiquimod does not.

Antigen-presenting cells expressing CD70, a member of the TNF superfamily engaging CD27 on T cells, are critical in CD8[+] T-cell responses (Brown *et al*, 1995; Sanchez *et al*, 2007). The superior potential of Langerin[+] dDC on CD8[+] T-cell priming (Henri *et al*, 2010) may be a consequence of their exclusive expression of CD70 (Elentner *et al*, 2009). In human LCs, which can present antigens to CD8[+] T cells (Cao *et al*, 2007; Klechevsky *et al*, 2010), there is similar evidence that CD70 expression correlates with the ability to induce proliferation of naïve (van der Aar *et al*, 2011) or memory (Polak *et al*, 2012) CD8[+] T cells.

The selective upregulation of CD70 by pIC/40 reflects qualitatively different DC activation states (Azukizawa *et al*, 2011; Shklovskaya *et al*, 2011) resulting from stimulation of different pattern-recognition receptors (PRRs). How such differential DC maturation programs establish *in vivo* remains a complex question, because danger signals are sensed and transmitted by a variety of immune and non-immune cells. Imiquimod engages TLR7 while poly(I:C) is sensed by TLR3 and cytoplasmic receptors RIG-I and MDA-5. Direct rather than bystander activation has been suggested to potentiate antigen presentation by DCs (Blander & Medzhitov, 2006). Unfortunately, few detailed studies of expression of PRRs and

response to their ligands are available for mouse skin DCs. Neither TLR3 nor TLR7 has been found on Langerin[+] dDCs or LCs so far (Fujita *et al*, 2004; Mitsui *et al*, 2004; Haley *et al*, 2012), whereas MDA-5 (Ifih1) and RIG-I (Ddx58) are detectable (Immunological Genome Project, http://www.immgen.org). Importantly, inflammatory signals may change the expression profile of PRRs by skin DCs.

Although human keratinocytes do express TLR7 (Flacher *et al*, 2006), there is no clear evidence that non-immune murine skin cells (fibroblasts, keratinocytes) respond to imiquimod and provide bystander activation signals to DCs (Drobits *et al*, 2012). However, dermal mast cells represent early responders for TLR7 stimulation, and they play a critical role in attracting pDCs and promoting LC emigration into lymph nodes (Heib *et al*, 2007;Drobits *et al*, 2012). Plasmacytoid DCs release high amounts of IFN-α (Palamara *et al*, 2004), which impairs maturation of LCs *in vitro* (Fujita *et al*, 2005). Since expression of type I IFN receptors on skin DCs remains unknown, the significance of this effect *in vivo* is not clear, but may explain why an antigen targeted to LCs is only poorly presented when the adjuvant is imiquimod.

Regarding poly(I:C), keratinocytes and fibroblasts express TLR3 (Drobits *et al*, 2012) and possibly cytoplasmic dsRNA receptors, resulting in the release of pro-inflammatory cytokines. This influence of poly(I:C) is probably potentiated by concomitant triggering of CD40 (Sanchez *et al*, 2007), providing a direct maturation signal to DCs.

Altogether, our observations demonstrate that despite a mostly similar apparent maturation, the influence of Langerin[+] dDCs and LCs on immune responses is essentially linked to the nature of the adjuvant. Elucidation of these mechanisms both in mouse and in human skin appears important and timely because DC-targeted vaccination with tumor antigens has now entered the clinical stage (Dhodapkar *et al*, 2014). We believe that systematic analysis of the sensitivity of skin DCs to different adjuvants now represents an important challenge that might help to reconcile previous contradictions and advance the recognition and application of skin DCs as a potent clinical target.

## Materials and Methods

### Mice

All mice used in this study were on a C57BL/6 genetic background. Inbred strains Ly5.1, C57BL/6, OT-I and OT-II were purchased from Charles River Laboratories (Sulzfeld, Germany). Langerin[−/−] mice (Kissenpfennig *et al*, 2005a) were a kind gift of Dr. Sem Saeland. Female mice were used for all experiments at 2–6 months of age. All experimental protocols were approved by the Austrian Federal Ministry of Science and Research, Department for Genetic Engineering and Animal Experimentation (#66.011/16-II/106/2008 and 66.011/0076-II/10b/2010).

### Reagents and antibodies

Cell cultures were performed in complete culture medium consisting of RPMI-1640 supplemented with 10% heat-inactivated FCS, 2 mM L-glutamine (Sigma, St. Louis, MO), 50 μg/ml gentamicin (PAA, Linz, Austria), and 50 μM beta-mercaptoethanol (Sigma).

Phenotypical analyses were performed by flow cytometry with mAb against CD4 (clone RM4-5), CD45.1 (clone A20), MHC class II (anti-I-A/I-E$^{diverse}$, clone 2G9), CD11c (clone HL3), CD8α (clone Ly-2), CD45 (clone 30-F11), CD103 (clone M290), EpCAM/CD326 (clone G8.8), CD19 (clone 1D3), NK1.1 (clone PK136), IL-12p40 (clone C15.6) (all from BD-Pharmingen, San Diego, CA), CD127 (clone A7R34, Biolegend, San Diego, CA), CD70 (clone FR70, Biolegend), and Langerin/CD207 mAb (clone 929F3; Dendritics, Lyon, France). When possible, viable cells were determined by exclusion of 7-AAD-positive dead cells (BD-Pharmingen). IL-12p40 and IL-10 stainings were performed on total lymph node cell suspension ($10^6$ cells/ml) incubated for 3 h in 1 μg/ml Brefeldin A (Golgiplug, BD-Pharmingen). To stain for Langerin or intracellular cytokines, permeabilization was performed with Cytofix/perm kit (BD-Pharmingen).

### Targeting antibodies

Ovalbumin-coupled anti-Langerin (Langerin/OVA), anti-DEC-205 (DEC/OVA) or isotype control (ISO/OVA) was produced at the Rockefeller University from antibody clones L31, NLDC145, and III/10, respectively (Idoyaga et al, 2008). All conjugates have a genetically engineered constant part which avoids binding to Fc receptors.

### Immunizations

Unless otherwise stated, 0.5 μg of Langerin/OVA, DEC/OVA, or ISO/OVA, diluted in 25 μl PBS (PAA, Linz, Austria), was injected intradermally into one or both ear pinna(e) of anesthetized mice, with or without adjuvant. For imiquimod treatment, approximately 40 μl of Aldara 5% imiquimod cream (a kind gift of Meda Pharma, Vienna, Austria), representing 2 mg imiquimod, was applied topically at each injection site. Alternatively, OVA-conjugated antibodies were injected i.d. together with 12.5 μg poly(I:C) (Sigma) and 12.5 μg anti-CD40 (clone 3/23, BD-Pharmingen).

In some experiments, intradermal injection of 0.5 μg of Langerin/OVA or ISO/OVA into one ear was followed by topical imiquimod treatment. One week later, 0.5 μg of DEC-OVA plus imiquimod was applied to the contralateral ear to trigger a cytotoxic response. Alternatively, both ears were immunized with Langerin/OVA or ISO/OVA in the presence of imiquimod, and the secondary response was initiated by i.p. injection of 2 μg DEC/OVA and 50 μg poly(I:C).

### Preparation of lymph node cell suspensions

Lymph nodes were harvested at the indicated times, and cell suspensions were obtained by digestion in PBS with 160 μg/ml Collagenase D and 120 μg/ml DNAse I (Roche Applied Science, Hamburg, Germany), for 25 min at 37°C.

### *In vivo* killing assays

At the indicated times after immunization, mice were injected i.v. with CD45.1⁺ cells, obtained from lymph nodes and spleen of Ly5.1 mice and differentially labeled with 20 or 200 nM CFSE (Invitrogen, Carlsbad, CA), and loaded with 10 or 100 nM OVA257–264 (OVA peptide SIINFEKL), respectively. As an internal control, unloaded cells labeled with 10 μM Cell-Tracker Orange (CTO; Invitrogen)

were mixed with CFSE-labeled cells. From each target cell population, we injected 3–6 × $10^6$ cells, meaning a total of 9–18 × $10^6$ target cells per mouse. Lymph nodes draining the immunization site and blood were collected 24 or 48 h after injection of target cells. Percentage of OVA-specific killing was calculated as described elsewhere (Hermans et al, 2004).

### T-cell transfer experiments

CD45.1⁺ ovalbumin-specific CD4⁺ and CD8⁺ T cells were obtained from F1 crossings of Ly5.1 mice with OT-II and OT-I mice, respectively. CD4⁺ or CD8⁺ T cells were purified by MACS separation (Miltenyi-Biotec, Bergisch Gladbach, Germany) from cell suspensions of lymphoid organs. Purified T cells were labeled with 0.5 μM CFSE for 10 min at room temperature, and $10^6$ OT-I or 5.$10^6$ OT-II was injected i.v. into congenic CD45.2⁺ C57BL/6 mice. The next day, mice were immunized as indicated. Six days later, we monitored by flow cytometry the proportion of T cells exhibiting diluted CFSE, indicating proliferation, and expressing IL-7R/CD127.

Three or 8 weeks after immunization, the percentage and absolute numbers of CD45.1⁺ CD8⁺ T cells in skin-draining lymph nodes were also evaluated. At these time points, total lymph node cells were cultured in the presence 1 μM OVA peptide SIINFEKL (Proimmune). After overnight incubation, lymph node cells were exposed to 1 μg/ml Brefeldin A (Golgiplug, BD-Pharmingen), before fixation, permeabilization, and flow cytometry analysis of IFN-γ production by individual CD45.1⁺ CD8⁺ T cells.

### Tracking of Langerin-targeted DC subsets

0.5 μg of PE-coupled full-length anti-Langerin L31 antibody (eBioscience, San Diego, CA) or rat IgG2a isotype control, diluted in 25 μL PBS, was injected intradermally into both ear pinnae of anesthetized mice, with or without adjuvants imiquimod or poly (I:C)+anti-CD40 (see Immunizations). Two or 4 days later, mice were sacrificed and auricular lymph nodes collected, digested, and analyzed by flow cytometry as described above.

### Irradiation and reconstitution by bone marrow grafts

C57BL/6, Ly5.1, or Langerin$^{-/-}$ mice were lethally irradiated with a single dose of 10 Gy. Immediately after irradiation, they were injected intravenously with 5 × $10^6$ bone marrow cells from C57BL/6, Ly5.1, or Langerin$^{-/-}$ mice. Eight to twelve weeks later, ear skin and skin-draining lymph nodes were obtained from chimera. In preliminary experiments, Ly5.1 mice expressing the congenic marker CD45.1 were used to obtain bone marrow or as a recipient for Langerin$^{-/-}$ bone marrow. In these chimeric mice, reconstitution of T- and B-cell populations was followed in the blood and lymph nodes and took at least 8 weeks. To characterize the reconstitution of Langerin⁺ skin DC subsets, epidermal sheets were stained with anti-Langerin hybridoma supernatant (clone 929F3), followed by chicken anti-rat immunoglobulin/Alexa 594 (Invitrogen), and counterstained with anti-MHCII/FITC (clone 2G9; BD-Pharmingen). Langerin expression was also determined by flow cytometry on CD45⁺ CD11c⁺ cells from digested epidermis. In cell suspensions from skin-draining lymph nodes, CD11c⁺ Langerin⁺ lymph node DCs were stained for CD103 and EpCAM.

## Endogenous ovalbumin-specific CD8$^+$ T cells

PE-coupled pentamers recognizing ovalbumin-specific CD8$^+$ T cells were purchased from Proimmune (Oxford, UK). Because the same mice had been injected with fluorescently labeled CD45.1$^+$ OVA-loaded and control target cells for simultaneous *in vivo* killing assays, ovalbumin-specific T cells were characterized as CD45.1$^-$ CD8$^+$ pentamer$^+$ from cell suspensions of skin-draining lymph nodes. CD19$^+$ B cells, NK1.1$^+$ NK/NKT cells, CD4$^+$ T cells and 7AAD$^+$ dead cells were also excluded.

## Tumor challenge

Mice were injected subcutaneously into the flank with 10$^5$ B16.OVA tumor cells (a kind gift of Dr. E.M. Lord and Dr. J.G. Frelinger, University of Rochester, Rochester, NY, USA (Lugade *et al*, 2005). Tumor size was assessed three times per week by measuring the short and long tumor diameters using calipers and is expressed as mean product of tumor volume (length × width$^2$). Four to five mice were used in each group. Measurements were stopped when one mouse in the cage reached maximum tumor size (> 1 cm in one of its dimensions) and had to be euthanized. The tumor sizes from all mice in each group were used to calculate the mean tumor size.

## Flow cytometry analyses

Experimental data were acquired on a FACSCalibur (Becton-Dickinson) and analyzed with the FlowJo software.

## Statistical tests

All experiments involved groups of at least three mice and were performed at least twice with similar results. Statistical analyses were performed using the GraphPad Prism software. Unpaired *t*-tests were used to compare two groups of data. For more than two groups, one-way ANOVA followed by post hoc Tukey's test was applied. Survival curves of tumor-bearing mice were compared using a log-rank Mantel–Cox test. *P*-values are indicated in the figure or in the corresponding legend, except in Tukey's tests for which GraphPad Prism only provides indicative values, that is, $P > 0.05$ (non-significant differences), $P < 0.05$ (*), $P < 0.01$ (**), and $P < 0.01$ (***). Error bars represent standard error of the mean.

**Supplementary information** for this article is available online: http://embomolmed.embopress.org

## Acknowledgements

This work was supported by the COMET Center ONCOTYROL (Project 2.3.1, Cell Therapy Unit), which is funded by the Austrian Federal Ministries for Transport, Innovation and Technology, and Economics, Family and Youth (via the Austrian Research Promotion Agency) and the Standortagentur Tirol. We appreciate the participation of the TILAK hospital holding company, who serves as a partner in the Oncotyrol research program. P.S. and D.G.M. are financed by the Austrian Science Fund (Grants FWF-P21487 and FWF-W1101 to P.S.). J.I. is supported by NIH/NIAMS grant 1K99AR062595. V.F. is supported by the Centre National pour la Recherche Scientifique and the Agence Nationale pour la Recherche (Program 'Investissements d'Avenir', ANR-11-EQPX-022). We thank Paul Eichberger and Prof. Peter Lukas from the Department of Therapeutic Radiology and Oncology at Innsbruck Medical University for their help with irradiating mice.

## Author contributions

VF, CHT and DGM performed the experiments. VF analyzed the results. RMS and JI provided key reagents and intellectual input. VF, PS, JI and NR designed the research. VF, PS and NR wrote the manuscript.

## Conflict of interest

RMS had financial interests in Celldex, which is developing targeting antibodies for human use. The other authors declare that they have no conflict of interest.

## References

van der Aar AM, de Groot R, Sanchez-Hernandez M, Taanman EW, van Lier RA, Teunissen MB, De Jong EC, Kapsenberg ML (2011) Cutting edge: virus selectively primes human langerhans cells for CD70 expression promoting CD8+ T cell responses. *J Immunol* 187: 3488–3492

## The paper explained

### Problem

Immunotherapy aims at specifically harnessing the immune system's potential to either dampen inflammatory responses or boost immunity. It is already employed in the clinics, for example with monoclonal antibodies that target receptors expressed by immune cells. In the near future, immunotherapy is expected to have a major impact for the treatment of conditions ranging from autoimmune diseases to cancer. Considerable efforts currently focus on targeting dendritic cells (DCs), which are instrumental for activation of T cells. We studied two distinct DC populations that inhabit the dermis (Langerin$^+$ dermal DCs) or the epidermis (Langerhans cells) of murine skin and express the endocytic receptor Langerin. Our goal was to determine how to manipulate antigen-specific killing by CD8$^+$ T cells through DCs *in vivo*, to generate either strong immunity or immune tolerance.

### Results

Conjugation of a model antigen, ovalbumin, to an antibody recognizing Langerin facilitates uptake by both DC subsets upon injection into the skin. Simultaneous application of one adjuvant combination including the TLR3 ligand poly(I:C) allowed potent stimulation of cytotoxic CD8$^+$ T cells. In contrast, imiquimod (TLR7 ligand) yielded poor primary cytotoxic responses against ovalbumin, and secondary (recall) responses were impaired, due to deletion of ovalbumin-specific T cells. Finally, in bone marrow chimera mice lacking Langerin exclusively in Langerhans cells, potent cytotoxic responses, but no tolerance induction, were observed. Importantly, this was achieved with either adjuvant, consistent with an intrinsic capacity of Langerhans cells to decrease immune responses.

### Impact

First, despite similar antigen presentation to CD8$^+$ T cells, Langerin$^+$ dermal DCs and Langerhans cells have distinct capacities to prime cytotoxic immune responses. Second, the use of distinct adjuvants allows to independently harness the different potential of simultaneously targeted DC subsets. Third, our results challenge the paradigm stating that DCs that sense danger signals via TLRs are inevitably prone to induce strong immune responses.

Azukizawa H, Dohler A, Kanazawa N, Nayak A, Lipp M, Malissen B, Autenrieth I, Katayama I, Riemann M, Weih F *et al* (2011) Steady state migratory RelB+ langerin+ dermal dendritic cells mediate peripheral induction of antigen-specific CD4+ CD25+ Foxp3+ regulatory T cells. *Eur J Immunol* 41: 1420–1434

Bedoui S, Whitney PG, Waithman J, Eidsmo L, Wakim L, Caminschi I, Allan RS, Wojtasiak M, Shortman K, Carbone FR *et al* (2009) Cross-presentation of viral and self antigens by skin-derived CD103+ dendritic cells. *Nat Immunol* 10: 488–495

Belz GT, Kallies A (2010) Effector and memory CD8+ T cell differentiation: toward a molecular understanding of fate determination. *Curr Opin Immunol* 22: 279–285

Blander JM, Medzhitov R (2006) Toll-dependent selection of microbial antigens for presentation by dendritic cells. *Nature* 440: 808–812

Bonifaz L, Bonnyay D, Mahnke K, Rivera M, Nussenzweig MC, Steinman RM (2002) Efficient targeting of protein antigen to the dendritic cell receptor DEC-205 in the steady state leads to antigen presentation on major histocompatibility complex class I products and peripheral CD8$^+$ T cell tolerance. *J Exp Med* 196: 1627–1638

Bonifaz LC, Bonnyay DP, Charalambous A, Darguste DI, Fujii SI, Soares H, Brimnes MK, Moltedo B, Moran TM, Steinman RM (2004) *In vivo* targeting of antigens to maturing dendritic cells via the DEC-205 receptor improves T cell vaccination. *J Exp Med* 199: 815–824

Boonstra A, Rajsbaum R, Holman M, Marques R, Asselin-Paturel C, Pereira JP, Bates EE, Akira S, Vieira P, Liu YJ *et al* (2006) Macrophages and myeloid dendritic cells, but not plasmacytoid dendritic cells, produce IL-10 in response to MyD88- and TRIF-dependent TLR signals, and TLR-independent signals. *J Immunol* 177: 7551–7558

Brown GR, Meek K, Nishioka Y, Thiele DL (1995) CD27-CD27 ligand/CD70 interactions enhance alloantigen-induced proliferation and cytolytic activity in CD8+ T lymphocytes. *J Immunol* 154: 3686–3695

Bursch LS, Rich BE, Hogquist KA (2009) Langerhans cells are not required for the CD8 T cell response to epidermal self-antigens. *J Immunol* 182: 4657–4664

Cao T, Ueno H, Glaser C, Fay JW, Palucka AK, Banchereau J (2007) Both Langerhans cells and interstitial DC cross-present melanoma antigens and efficiently activate antigen-specific CTL. *Eur J Immunol* 37: 2657–2667

Chatterjee B, Smed-Sorensen A, Cohn L, Chalouni C, Vandlen R, Lee BC, Widger J, Keler T, Delamarre L, Mellman I (2012) Internalization and endosomal degradation of receptor-bound antigens regulate the efficiency of cross presentation by human dendritic cells. *Blood* 120: 2011–2020

Chen L (2004) Co-inhibitory molecules of the B7-CD28 family in the control of T-cell immunity. *Nat Rev Immunol* 4: 336–347

Choi JH, Do Y, Cheong C, Koh H, Boscardin SB, Oh YS, Bozzacco L, Trumpfheller C, Park CG, Steinman RM (2009) Identification of antigen-presenting dendritic cells in mouse aorta and cardiac valves. *J Exp Med* 206: 497–505

Clynes RA, Towers TL, Presta LG, Ravetch JV (2000) Inhibitory Fc receptors modulate *in vivo* cytotoxicity against tumor targets. *Nat Med* 6: 443–446

Cohn L, Chatterjee B, Esselborn F, Smed-Sorensen A, Nakamura N, Chalouni C, Lee BC, Vandlen R, Keler T, Lauer P *et al* (2013) Antigen delivery to early endosomes eliminates the superiority of human blood BDCA3+ dendritic cells at cross presentation. *J Exp Med* 210: 1049–1063

Delamarre L, Holcombe H, Mellman I (2003) Presentation of exogenous antigens on major histocompatibility complex (MHC) class I and MHC class II molecules is differentially regulated during dendritic cell maturation. *J Exp Med* 198: 111–122

Dhodapkar MV, Sznol M, Zhao B, Wang D, Carvajal RD, Keohan ML, Chuang E, Sanborn RE, Lutzky J, Powderly J *et al* (2014) Induction of antigen-specific immunity with a vaccine targeting NY-ESO-1 to the dendritic cell receptor DEC-205. *Sci Transl Med* 6: 232ra51

Dickgreber N, Stoitzner P, Bai Y, Price KM, Farrand KJ, Manning K, Angel CE, Dunbar PR, Ronchese F, Fraser JD *et al* (2009) Targeting antigen to MHC class II molecules promotes efficient cross-presentation and enhances immunotherapy. *J Immunol*, 182: 1260–12699

Elentner A, Finke D, Schmuth M, Chappaz S, Ebner S, Malissen B, Kissenpfennig A, Romani N, Dubrac S (2009) Langerhans cells are critical in the development of atopic dermatitis-like inflammation and symptoms in mice. *J Cell Mol Med* 13: 2658–2672

Flacher V, Bouschbacher M, Verronèse E, Massacrier C, Berthier-Vergnes O, De Saint-Vis B, Caux C, Dezutter-Dambuyant C, Lebecque S, Valladeau J (2006) Human Langerhans cells express a specific TLR profile and differentially respond to viruses and Gram-positive bacteria. *J Immunol* 177: 7959–7967

Flacher V, Douillard P, Ait-Yahia S, Stoitzner P, Clair-Moninot V, Romani N, Saeland S (2008) Expression of langerin/CD207 reveals dendritic cell heterogeneity between inbred mouse strains. *Immunology* 123: 339–347

Flacher V, Tripp CH, Stoitzner P, Haid B, Ebner S, Del Frari B, Koch F, Park CG, Steinman RM, Idoyaga J *et al* (2010) Epidermal Langerhans cells rapidly capture and present antigens from C-type lectin-targeting antibodies deposited in the dermis. *J Invest Dermatol* 130: 755–762

Flacher V, Tripp CH, Haid B, Kissenpfennig A, Malissen B, Stoitzner P, Idoyaga J, Romani N (2012) Skin langerin+ dendritic cells transport intradermally injected anti-DEC-205 antibodies but are not essential for subsequent cytotoxic CD8+ T cell responses. *J Immunol* 188: 2146–2155

Fujita H, Asahina A, Mitsui H, Tamaki K (2004) Langerhans cells exhibit low responsiveness to double-stranded RNA. *Biochem Biophys Res Commun* 319: 832–839

Fujita H, Asahina A, Tada Y, Fujiwara H, Tamaki K (2005) Type I interferons inhibit maturation and activation of mouse Langerhans cells. *J Invest Dermatol* 125: 126–133

Galon J, Costes A, Sanchez-Cabo F, Kirilovsky A, Mlecnik B, Lagorce-Pages C, Tosolini M, Camus M, Berger A, Wind P *et al* (2006) Type, density, and location of immune cells within human colorectal tumors predict clinical outcome. *Science* 313: 1960–1964

Gill D, Tan PH (2010) Induction of pathogenic cytotoxic T lymphocyte tolerance by dendritic cells: a novel therapeutic target. *Expert Opin Ther Targets* 14: 797–824

Ginhoux F, Tacke F, Angeli V, Bogunovic M, Loubeau M, Dai XM, Stanley ER, Randolph GJ, Merad M (2006) Langerhans cells arise from monocytes *in vivo*. *Nat Immunol* 7: 265–273

Ginhoux F, Collin MP, Bogunovic M, Abel M, Leboeuf M, Helft J, Ochando J, Kissenpfennig A, Malissen B, Grisotto M *et al* (2007) Blood-derived dermal langerin+ dendritic cells survey the skin in the steady state. *J Exp Med* 204: 3133–3146

Gomez de Agüero M, Vocanson M, Hacini-Rachinel F, Taillardet M, Sparwasser T, Kissenpfennig A, Malissen B, Kaiserlian D, Dubois B (2012) Langerhans cells protect from allergic contact dermatitis in mice by tolerizing CD8+ T cells and activating Foxp3+ regulatory T cells. *J Clin Invest* 122: 1700–1711

Haley K, Igyarto BZ, Ortner D, Bobr A, Kashem S, Schenten D, Kaplan DH (2012) Langerhans cells require MyD88-dependent signals for *Candida albicans* response but not for contact hypersensitivity or migration. *J Immunol* 188: 4334–4339

Hawiger D, Inaba K, Dorsett Y, Guo M, Mahnke K, Rivera M, Ravetch JV, Steinman RM, Nussenzweig MC (2001) Dendritic cells induce peripheral T cell unresponsiveness under steady state conditions *in vivo*. *J Exp Med* 194: 769–779

Henri S, Poulin LF, Tamoutounour S, Ardouin L, Guilliams M, de Bovis B, Devilard E, Viret C, Azukizawa H, Kissenpfennig A *et al* (2010) CD207+ CD103+ dermal dendritic cells cross-present keratinocyte-derived antigens irrespective of the presence of Langerhans cells. *J Exp Med* 207: 189–206

Hermans IF, Silk JD, Yang J, Palmowski MJ, Gileadi U, McCarthy C, Salio M, Ronchese F, Cerundolo V (2004) The VITAL assay: a versatile fluorometric technique for assessing CTL- and NKT-mediated cytotoxicity against multiple targets *in vitro* and *in vivo*. *J Immunol Methods* 285: 25–40

Holcmann M, Stoitzner P, Drobits B, Luehrs P, Stingl G, Romani N, Maurer D, Sibilia M (2009) Skin inflammation is not sufficient to break tolerance induced against a novel antigen. *J Immunol* 183: 1133–1143

Idoyaga J, Cheong C, Suda K, Suda N, Kim JY, Lee H, Park CG, Steinman RM (2008) Cutting edge: langerin/CD207 Receptor on dendritic cells mediates efficient antigen presentation on MHC I and II products *in vivo*. *J Immunol* 180: 3647–3650

Idoyaga J, Suda N, Suda K, Park CG, Steinman RM (2009) Antibody to Langerin/CD207 localizes large numbers of CD8alpha+ dendritic cells to the marginal zone of mouse spleen. *Proc Natl Acad Sci USA* 106: 1524–1529

Idoyaga J, Lubkin A, Fiorese C, Lahoud MH, Caminschi I, Huang Y, Rodriguez A, Clausen BE, Park CG, Trumpfheller C *et al* (2011) Comparable T helper 1 (Th1) and CD8 T-cell immunity by targeting HIV gag p24 to CD8 dendritic cells within antibodies to Langerin, DEC205, and Clec9A. *Proc Natl Acad Sci USA* 108: 2384–2389

Idoyaga J, Fiorese C, Zbytnuik L, Lubkin A, Miller J, Malissen B, Mucida D, Merad M, Steinman RM (2013) Specialized role of migratory dendritic cells in peripheral tolerance induction. *J Clin Invest* 123: 844–854

Igyarto BZ, Jenison MC, Dudda JC, Roers A, Muller W, Koni PA, Campbell DJ, Shlomchik MJ, Kaplan DH (2009) Langerhans cells suppress contact hypersensitivity responses via cognate CD4 interaction and langerhans cell-derived IL-10. *J Immunol* 183: 5085–5093

Igyarto BZ, Haley K, Ortner D, Bobr A, Gerami-Nejad M, Edelson BT, Zurawski SM, Malissen B, Zurawski G, Berman J *et al* (2011) Skin-resident murine dendritic cell subsets promote distinct and opposing antigen-specific T helper cell responses. *Immunity* 35: 260–272

Jiang A, Bloom O, Ono S, Cui W, Unternaehrer J, Jiang S, Whitney JA, Connolly J, Bancheerau J, Mellman I (2007) Disruption of E-cadherin-mediated adhesion induces a functionally distinct pathway of dendritic cell maturation. *Immunity* 27: 610–624

Joffre O, Nolte MA, Sporri R, Reis e Sousa C (2009) Inflammatory signals in dendritic cell activation and the induction of adaptive immunity. *Immunol Rev* 227: 234–247

Joffre OP, Segura E, Savina A, Amigorena S (2012) Cross-presentation by dendritic cells. *Nat Rev Immunol* 12: 557–569

Kissenpfennig A, Aït-Yahia S, Clair-Moninot V, Stössel H, Badell E, Bordat Y, Pooley JL, Lang T, Prina E, Coste I *et al* (2005a) Disruption of the *langerin/ CD207* gene abolishes Birbeck granules without a marked loss of Langerhans cell function. *Mol Cell Biol* 25: 88–99

Kissenpfennig A, Henri S, Dubois B, Laplace-Builhé C, Perrin P, Romani N, Tripp CH, Douillard P, Leserman L, Kaiserlian D *et al* (2005b) Dynamics and function of Langerhans cells *in vivo*: dermal dendritic cells colonize lymph node areas distinct from slower migrating Langerhans cells. *Immunity* 22: 643–654

Klechevsky E, Flamar AL, Cao Y, Blanck JP, Liu M, O'Bar A, Agouna-Deciat O, Klucar P, Thompson-Snipes L, Zurawski S *et al* (2010) Cross-priming CD8+ T cells by targeting antigens to human dendritic cells through DCIR. *Blood* 116: 1685–1697

Kurts C, Kosaka H, Carbone FR, Miller JFAP, Heath WR (1997) Class I-restricted cross-presentation of exogenous self- antigens leads to deletion of autoreactive CD8[+] T cells. *J Exp Med* 186: 239–245

Liu YJ (2007) Thymic stromal lymphopoietin and OX40 ligand pathway in the initiation of dendritic cell-mediated allergic inflammation. *J Allergy Clin Immunol* 120: 238–244

Lugade AA, Moran JP, Gerber SA, Rose RC, Frelinger JG, Lord EM (2005) Local radiation therapy of B16 melanoma tumors increases the generation of tumor antigen-specific effector cells that traffic to the tumor. *J Immunol* 174: 7516–7523

Lutz MB, Schuler G (2002) Immature, semi-mature and fully mature dendritic cells: which signals induce tolerance or immunity? *Trends Immunol* 23: 445–449

Lutz MB, Kurts C (2009) Induction of peripheral CD4+ T-cell tolerance and CD8+ T-cell cross-tolerance by dendritic cells. *Eur J Immunol* 39: 2325–2330

Mahnke K, Guo M, Lee S, Sepulveda H, Swain SL, Nussenzweig M, Steinman RM (2000) The dendritic cell receptor for endocytosis, DEC-205, can recycle and enhance antigen presentation via major histocompatibility complex class II-positive lysosomal compartments. *J Cell Biol* 151: 673–683

Mellman I, Coukos G, Dranoff G (2011) Cancer immunotherapy comes of age. *Nature* 480: 480–489

Merad M, Manz MG, Karsunky H, Wagers A, Peters W, Charo I, Weissman IL, Cyster JG, Engleman EG (2002) Langerhans cells renew in the skin throughout life under steady-state conditions. *Nat Immunol* 3: 1135–1141

Mitsui H, Watanabe T, Saeki H, Mori K, Fujita H, Tada Y, Asahina A, Nakamura K, Tamaki K (2004) Differential expression and function of Toll-like receptors in Langerhans cells: comparison with splenic dendritic cells. *J Invest Dermatol* 122: 95–102

Nagao K, Kobayashi T, Moro K, Ohyama M, Adachi T, Kitashima DY, Ueha S, Horiuchi K, Tanizaki H, Kabashima K *et al* (2012) Stress-induced production of chemokines by hair follicles regulates the trafficking of dendritic cells in skin. *Nat Immunol* 13: 744–752

Noordegraaf M, Flacher V, Stoitzner P, Clausen BE (2010) Functional redundancy of Langerhans cells and Langerin+ dermal dendritic cells in contact hypersensitivity. *J Invest Dermatol* 130: 2752–2759

Palamara F, Meindl S, Holcmann M, Luhrs P, Stingl G, Sibilia M (2004) Identification and characterization of pDC-like cells in normal mouse skin and melanomas treated with imiquimod. *J Immunol* 173: 3051–3061

Palucka K, Bancheerau J, Mellman I (2010) Designing vaccines based on biology of human dendritic cell subsets. *Immunity* 33: 464–478

Platt CD, Ma JK, Chalouni C, Ebersold M, Bou-Reslan H, Carano RA, Mellman I, Delamarre L (2010) Mature dendritic cells use endocytic receptors to capture and present antigens. *Proc Natl Acad Sci USA* 107: 4287–4292

Polak ME, Newell L, Taraban VY, Pickard C, Healy E, Friedmann PS, Al-Shamkhani A, Ardern-Jones MR (2012) CD70-CD27 interaction augments CD8+ T-cell activation by human epidermal Langerhans cells. *J Invest Dermatol* 132: 1636–1644

Pooley JL, Heath WR, Shortman K (2001) Cutting edge: intravenous soluble antigen is presented to CD4 T cells by CD8[−] dendritic cells, but cross-presented to CD8 T cells by CD8[+] dendritic cells. *J Immunol* 166: 5327–5330

Poulin LF, Henri S, de Bovis B, Devilard E, Kissenpfennig A, Malissen B (2007) The dermis contains langerin+ dendritic cells that develop and function independently of epidermal Langerhans cells. *J Exp Med* 204: 3119–3131

Rechtsteiner G, Warger T, Osterloh P, Schild H, Radsack MP (2005) Cutting edge: priming of CTL by transcutaneous peptide immunization with imiquimod. *J Immunol* 174: 2476–2480

Sanchez PJ, McWilliams JA, Haluszczak C, Yagita H, Kedl RM (2007) Combined TLR/CD40 stimulation mediates potent cellular immunity by regulating dendritic cell expression of CD70 *in vivo*. *J Immunol* 178: 1564–1572

Sancho D, Mourao-Sa D, Joffre OP, Schulz O, Rogers NC, Pennington DJ, Carlyle JR, Reis e Sousa C (2008) Tumor therapy in mice via antigen targeting to a novel, DC-restricted C-type lectin. *J Clin Invest* 118: 2098–2110

Sancho D, Reis e Sousa C (2012) Signaling by myeloid C-type lectin receptors in immunity and homeostasis. *Annu Rev Immunol* 30: 491–529

Segura E, Durand M, Amigorena S (2013) Similar antigen cross-presentation capacity and phagocytic functions in all freshly isolated human lymphoid organ-resident dendritic cells. *J Exp Med* 210: 1035–1047

Shklovskaya E, O'Sullivan BJ, Ng LG, Roediger B, Thomas R, Weninger W, Fazekas de St Groth B (2011) Langerhans cells are precommitted to immune tolerance induction. *Proc Natl Acad Sci USA* 108: 18049–18054

Soares H, Waechter H, Glaichenhaus N, Mougneau E, Yagita H, Mizenina O, Dudziak D, Nussenzweig MC, Steinman RM (2007) A subset of dendritic cells induces CD4+ T cells to produce IFN-{gamma} by an IL-12-independent but CD70-dependent mechanism *in vivo*. *J Exp Med* 204: 1095–1106

Spörri R, Reis e Sousa C (2005) Inflammatory mediators are insufficient for full dendritic cell activation and promote expansion of CD4+ T cell populations lacking helper function. *Nat Immunol* 6: 163–170

Steinman RM (2012) Decisions about dendritic cells: past, present, and future. *Annu Rev Immunol* 30: 1–22

Stoitzner P, Tripp CH, Eberhart A, Price KM, Jung JY, Bursch LS, Ronchese F, Romani N (2006) Langerhans cells cross-present antigen derived from skin. *Proc Natl Acad Sci USA* 103: 7783–7788

Suzuki H, Wang BH, Shivji GM, Toto P, Amerio P, Tomai MA, Miller RL, Sauder DN (2000) Imiquimod, a topical immune response modifier, induces migration of Langerhans cells. *J Invest Dermatol* 114: 135–141

Tacken PJ, de Vries IJ, Torensma R, Figdor CG (2007) Dendritic-cell immunotherapy: from *ex vivo* loading to *in vivo* targeting. *Nat Rev Immunol* 7: 790–802

Trinchieri G (2003) Interleukin-12 and the regulation of innate resistance and adaptive immunity. *Nat Rev Immunol* 3: 133–146

Trumpfheller C, Caskey M, Nchinda G, Longhi MP, Mizenina O, Huang Y, Schlesinger SJ, Colonna M, Steinman RM (2008) The microbial mimic poly IC induces durable and protective CD4+ T cell immunity together with a dendritic cell targeted vaccine. *Proc Natl Acad Sci USA* 105: 2574–2579

Van der Fits L, Mourits S, Voerman JS, Kant M, Boon L, Laman JD, Cornelissen F, Mus AM, Florencia E, Prens EP *et al* (2009) Imiquimod-induced psoriasis-like skin inflammation in mice is mediated via the IL-23/IL-17 axis. *J Immunol* 182: 5836–5845

Wohn C, Ober-Blobaum JL, Haak S, Pantelyushin S, Cheong C, Zahner SP, Onderwater S, Kant M, Weighardt H, Holzmann B *et al* (2013) Langerin (neg) conventional dendritic cells produce IL-23 to drive psoriatic plaque formation in mice. *Proc Natl Acad Sci USA* 110: 10723–10728

Yamazaki S, Inaba K, Tarbell KV, Steinman RM (2006) Dendritic cells expand antigen-specific Foxp3+ CD25+ CD4+ regulatory T cells including suppressors of alloreactivity. *Immunol Rev* 212: 314–329

Yamazaki S, Dudziak D, Heidkamp GF, Fiorese C, Bonito AJ, Inaba K, Nussenzweig MC, Steinman RM (2008) CD8+ CD205+ splenic dendritic cells are specialized to induce Foxp3+ regulatory T cells. *J Immunol* 181: 6923–6933

