## [Review Process File · EMBO Molecular Medicine]

Murine Langerin⁺ dermal dendritic cells prime CD8⁺ T cells while Langerhans cells induce cross-tolerance.

Vincent Flacher, Christoph H. Tripp, David G. Mairhofer, Ralph M. Steinman, Patrizia Stoitzner, Juliana Idoyaga, and Nikolaus Romani

Corresponding authors: Nikolaus Romani and Patrizia Stoitzner, Innsbruck Medical University

Review timeline:

Submission date:	12 July 2013
Editorial Decision:	10 September 2013
Revision received:	07 May 2014
Editorial Decision:	26 June 2014
Revision received:	04 July 2014
Accepted:	08 July 2014

Transaction Report:

Editor: Céline Carret

1st Editorial Decision

10 September 2013

Thank you for the submission of your manuscript to EMBO Molecular Medicine. I am sorry that it has taken so long to get back to you on your manuscript.

While reviewers 1 and 3 delivered their evaluations in a timely manner, we did not receive the other reviewers' input. As the evaluations from the first two reviewers are consistent, and a further delay cannot be justified, I have decided to proceed based on these evaluations.

You will see that while both reviewers are generally supportive of your work and underline its potential interest, they also raise a number of specific concerns that should be addressed in a major revision of the current manuscript.

As you will see from the enclosed reports, both referees suggest strengthening the paper by experimentally characterising the underlying mechanism of TLR7 agonist treatment and induction of tolerance. In addition, you will see that both referees highlight specific points that should also be considered.

As we feel that the suggested revisions are reasonable and would considerably improve the manuscript, we would encourage you to address all issues as best as possible. Please note that it is EMBO Molecular Medicine policy to allow only a single round of major revision and that, as acceptance or rejection of the manuscript will depend on another round of review, your responses should be as complete as possible.

Please take this opportunity to reformat the article according to EMBO Mol. Med. guidelines: [http://onlinelibrary.wiley.com/journal/10.1002/\(ISSN\)1757-4684/homepage/ForAuthors.html](http://onlinelibrary.wiley.com/journal/10.1002/(ISSN)1757-4684/homepage/ForAuthors.html)

I look forward to seeing a revised form of your manuscript as soon as possible.

***** Reviewer's comments *****

Referee #1 (Comments on Novelty/Model System):

"Langerin+ dermal dendritic cells prime CD8+ T cells while Langerhans cells induce cross-tolerance."

Flacher et al. present data describing the elicitation of cytotoxic CD8 T cells via immunization with antigen (OVA) coupled to anti-Langerin antibody. Targeting antigen to the Langerin, without additional adjuvants failed to induce CD8 T cells. The right combination of a TLR agonist, i.e. pIC with anti-CD40 was a good inducer of CD8 T cells which were protective in a tumor-challenge model. Interestingly, the combination of the TLR7 agonist imiquimod (the wrong TLR agonist) and OVA-anti-Langerin induced tolerance to vaccination. Such tolerance was mediated by the LC population.

Overall impressions:

- iAuthors compile results from multiple experiments to perform statistical analysis. I prefer representative experiments as pooling results can be misleading.
- iThe authors do not compare parameters induced by the anti-Langerin/OVA to soluble OVA. This makes interpretation of the results difficult.
- iWhile interesting phenomena, the link between TLR7 agonist treatment and induction tolerance with the anti-Langerin/OVA conjugate is not characterized.

major points

1. anti-CD40 and poly(i:c) with soluble OVA is a potent inducer of cytotoxic CD8 T cells. How does the anti-Langerin/OVA compare to soluble OVA?
2. The authors should demonstrate that the doses of L31 used lead to specific targeting of the antigen to Langerin expressing cells. It may be that the anti-CD40/poly (i:c) affects the distribution of antigen as well as the cells and this leads to enhanced vaccination.
3. Within Figure 3, the authors stipulate that there were no differences in the induction of a number of surface markers or IL-10 within the Langerin+dDCs or the LCs. This data should be shown, and staining should be confirmed with positive controls or shown to increase when compared to the no-treatment group, as flow-panel design alone can cause detection failure.
4. While an interesting phenotype, the induction of tolerance by anti-Langerin/OVA/imiquimod, as it stands, is mechanistically unclear. Is this induction a product CD8 extrinsic or intrinsic mechanisms? Adoptive transfer systems could be used to evaluate if the environment or the CD8 T cells themselves are affected.

minor points

1. OVA systems carry the caveat that OTI cells are extremely sensitive to antigen stimulation. Have the authors examined response to other protein/antibody conjugates?
2. The authors do not indicate the numbers of OTI cells that were transferred within adoptive transfer assays. This is important due to the sensitivity of responses to the numbers of founder cells.
3. Could this same system of tolerance induction be applied to demonstrate benefit, e.g. prevention allergy?
4. The running title should be reworded: "CD8+ T cells control by Langerin+ skin DCs"
5. The statement within the introduction "Once activated, DCs migrate to lymph nodes and prime antigen-specific T cells there. Thus, DCs are essential to initiate adaptive immune responses (Palucka et al., 2010;Steinman, 2012)" does not support the conclusion.
6. With the introduction the rather with the text should be removed: "However, since imiquimod triggers emigration and maturation of skin DCs, including LCs (Flacher et al., 2008;Suzuki et al., 2000), its tolerogenic influence appears to rather support alternatives to this paradigm (Joffre et al., 2009)

Referee #3 (Comments on Novelty/Model System):

After the addition of some new data and some better or more thorough explanations, this paper

should definitely be published.

Referee #3 (Remarks):

Dendritic cells (DC) are a heterogeneous population of antigen presenting cells that can induce immunity or tolerance depending on the context in which a given antigen is captured. DCs found residing under steady state conditions, or ones that have been activated (matured) by non-inflammatory signals, are generally tolerogenic, although critical unknowns remain regarding what distinguishes DCs that induce tolerance from those that induce immunity. The issue remains a fundamental unknown in our understanding of the immune response and involves problems of both immunology and cell biology, meaning that it is an issue of general interest. In this manuscript, the authors have extended their previous work on the functional characterization of murine skin DCs. They deliver antigen through the surface lectin langerin to specifically target Langerhans cells (LCs) and dermal DCs *in vivo*. The data show that dermal DCs are the major DC subset contributing to CD8 T cell priming by Langerin-targeted ovalbumin when injected with the adjuvants polyIC + anti-CD40, whereas LCs do not prime immunity. The data further show that langerin-targeted ovalbumin in combination with a TLR7 agonist (imiquimod) induces tolerance, and that LCs are required for induction of tolerance. This is surprising since imiquimod has previously been shown to induce CD8 T cell immunity and may have important implications as imiquimod is currently used in patients to stimulate immune responses. These results also suggest that TLR7 agonists, and maybe other microbial products, may have either stimulatory or tolerogenic effects depending on the DC subset involved. This is an interesting study that could be strengthened by better characterizing the mechanism underlying these observations. It would be important for the authors to further characterize or at least explain how imiquimod can induce either CD8 T cell immunity or tolerance depending on the context.

Although the authors here show that imiquimod combined with intradermal injection of Langerin-targeted ovalbumin induced tolerance, in a previous report (Flacher et al, 2012), imiquimod combined with intradermal injection of DEC205-targeted ovalbumin induced CD8 immunity. Langerhans cells were targeted in both vaccination schemes since they express both Langerin and DEC205. It could be that level of antigen presentation by LCs is different following antigen uptake through DEC205 and langerin and determines the capacity of LCs to induce tolerance or immunity. Alternatively, DEC205 could also target other immunostimulatory DC subsets that outcompete LC tolerogenic activity. Another interesting question not addressed in this study is how LCs and dermal DCs are matured since neither express TLR7 and TLR3 and are thus not directly matured by imiquimod and polyIC. The authors do refer to this issue in the text, but additional information or discussion is needed to explain or rationalize the mechanism involved.

Additionally, it has been previously shown that steady-state DCs induce tolerance. Yet here, langerin-targeted OVA alone induces T cell proliferation and exhibits a capacity to induce memory T cells similar to Langerin-targeted OVA combined imiquimod (Figure 1B). I thus found it surprising that langerin-targeted OVA alone did not induce tolerance (Figure 5C).

In Figure 4A the authors found that LCs did not contribute to CD8 T cell immunity upon immunization with langerin-targeted OVA combined polyIC and anti-CD40. However, they did not explore whether they could also be tolerogenic. This could be addressed by priming LKO->WT mice with langerin-targeted OVA combined with polyIC followed by a boost with DEC-OVA combined with polyIC.

Figure 4A shows that LCs targeted alone in presence of PolyIC and anti-CD40 did not induce cytotoxic T cells. Nevertheless, the authors were still able to induce proliferation of memory IFN γ + CD8 T cells to the same degree as WT mice. The authors should comment on these results, which appear inconsistent.

Minor point:

In the introduction the authors mention that expression of langerin is heterogeneous between mouse strains (Flacher, 2008). It is unclear which DC subsets express Langerin in the mouse strain used in this study and should be clarified.

POINT-BY-POINT REPLY TO THE REVIEWERS

May 7th 2014

Referee #1 (Comments on Novelty/Model System):

"Langerin+ dermal dendritic cells prime CD8+ T cells while Langerhans cells induce cross-tolerance."

Flacher et al. present data describing the elicitation of cytotoxic CD8 T cells via immunization with antigen (OVA) coupled to anti-Langerin antibody. Targeting antigen to the Langerin, without additional adjuvants failed to induce CD8 T cells. The right combination of a TLR agonist, i.e. pIC with anti-CD40 was a good inducer of CD8 T cells which were protective in a tumor-challenge model. Interestingly, the combination of the TLR7 agonist imiquimod (the wrong TLR agonist) and OVA-anti-Langerin induced tolerance to vaccination. Such tolerance was mediated by the LC population.

General impressions

#1: Authors compile results from multiple experiments to perform statistical analysis. I prefer representative experiments as pooling results can be misleading.

AUTHORS' RESPONSE: We fully understand the reviewer's concern. For instance, we do believe that individual, representative flow cytometry profiles should always be available to the reader in addition to graphs compiling fluorescence intensity or percentages. In fact, we do have representative experiments plus the graphs on several occasions throughout the manuscript (Figure 1 - 3; Fig. E1, E3). However, our individual experiments were typically performed with ~3 mice per condition, which is not sufficient to perform reliable statistical analysis without repetition. Because of this, we believe that the best way to plot graphs and to determine the statistical significance of our data is to pool experimental points.

#2: The authors do not compare parameters induced by the anti-Langerin/OVA to soluble OVA. This makes interpretation of the results difficult.

AUTHORS' RESPONSE: Antigen uptake and processing is facilitated by coupling to antibodies specific to DCs. There is a large body of published evidence for this concept showing that targeted antigens elicit much higher lymphocyte responses as compared with soluble proteins / vaccines. (reviewed in e.g., Caminschi et al., 2012; Kreutz et al., 2013; Steinman, 2012; Trumpheller et al., 2011). Therefore, we consider that OVA alone is not a relevant control in our series of experiments (please see further details in our response to Major point #1).

#3: While interesting phenomena, the link between TLR7 agonist treatment and induction tolerance with the anti-Langerin/OVA conjugate is not characterized.

AUTHORS' RESPONSE: We did not aim to demonstrate a specific link between TLR7 agonists and tolerance, but rather that two adjuvants triggering apparent maturation of DCs, namely imiquimod and poly(I:C)+anti-CD40, are not entirely equivalent in terms of immune response induction. We observed different profiles of expression of DC receptors (i.e. CD70 in Langerin+ dDCs but not in LCs) associated with these strong or abortive CD8+ T cell responses.

major points

1. anti-CD40 and poly(i:c) with soluble OVA is a potent inducer of cytotoxic CD8 T cells. How does the anti-Langerin/OVA compare to soluble OVA?

AUTHORS' RESPONSE: We agree that non-DC-specific uptake of soluble OVA may result in a different outcome than OVA/antibody conjugates. However, side-by-side comparison with soluble OVA is not likely to bring further insight into the mechanism that underlies tolerance induction in our experimental settings. Indeed, it has been well established that antigen uptake and processing is greatly facilitated by coupling OVA to antibodies specific to DCs, and that it relies on different intracellular pathways. As a consequence, the amount of OVA antigen necessary to immunize a mouse can be lowered by several orders of magnitude (>1000 times for aDEC/OVA vs. OVA) (Bonifaz et al., 2004). Further work showed a similar potential for aDEC/OVA and Langerin/OVA (Idoyaga et al., 2008; Idoyaga et al., 2011). For these reasons, we have preferred to use ISO/OVA (modified, like Langerin/OVA, to avoid Fc receptor-mediated uptake) as a negative control in our experiments.

2. The authors should demonstrate that the doses of L31 used lead to specific targeting of the antigen to Langerin expressing cells. It may be that the anti-CD40/poly (i:c) affects the distribution of antigen as well as the cells and this leads to enhanced vaccination.

*AUTHORS' RESPONSE: Upon injection into the skin, we have previously demonstrated binding specificity of the anti-Langerin L31 clone to Langerin+ dermal DCs, LCs and, in mice with BALB/c genetic background, CD8+ DCs (Flacher et al., 2012; Idoyaga et al., 2009). However, inflammatory stimuli may affect antigen distribution to DC subsets, for instance by inducing Langerin expression in potentially cross-presenting LN-resident CD8+ DCs, which normally lack Langerin in C57BL/6 mice (Flacher et al., 2008). Using imiquimod as an adjuvant, we did previously not observe an up-regulation of intracellular Langerin (clone 929F3) in the LN-resident DC population – neither in the here used C57BL/6 nor in BALB/c mice (Flacher et al., 2008). To address the reviewer's concern, we have extended these analyses and used a full-length anti-Langerin L31 antibody directly coupled to PE to verify whether our second adjuvant, namely polyI:C plus anti-CD40, leads to the targeting of different DC subsets. We did not find any targeting of CCR7- CD8+ LN-resident DCs, as shown now in the new **Figure E2A**. On the other hand, we did observe some targeting of skin-derived Langerin-negative dDCs. However, we believe this population plays only a negligible role, if any, in CD8 T cell priming because these Langerin-negative dDCs captured not only anti-Langerin/PE antibodies, but also isotype control antibodies. This strongly suggests that we observe an Fc Receptor-dependent uptake. This cannot occur with OVA conjugates, because of a mutation introduced of their FcR-binding-domain (Clynes et al., 2000; Hawiger et al., 2001).*

3. Within Figure 3, the authors stipulate that there were no differences in the induction of a number of surface markers or IL-10 within the Langerin+dDCs or the LCs. This data should be shown, and staining should be confirmed with positive controls or shown to increase when compared to the no-treatment group, as flow-panel design alone can cause detection failure.

*AUTHORS' RESPONSE: As asked by the reviewer, our new **Figure E3A** shows the expression profile of PD-L1, PD-L2 and ICOSL in Langerin+ skin-derived DCs in LNs following treatment with imiquimod or pIC/40. Additional analyses are depicted in **Figure E3B** that show the absence of IL-10 production and OX-40 ligand expression by any DC subset. Our flow cytometry*

plots include isotype controls as well as untreated control mice. Unfortunately, we could not provide a positive control for this staining, because, despite repeated attempts, we were unable to set up experimental conditions triggering IL-10 production in LCs or Langerin+ dDCs from skin-draining LNs.

4. While an interesting phenotype, the induction of tolerance by anti-Langerin/OVA/imiquimod, as it stands, is mechanistically unclear. Is this induction a product CD8 extrinsic or intrinsic mechanisms? Adoptive transfer systems could be used to evaluate if the environment or the CD8 T cells themselves are affected.

AUTHORS' RESPONSE: We apologize for not having stated clearly enough in the original manuscript, that the decrease in cell numbers of either endogenous (Figure 6) or transgenic (Figure 2) OVA-specific CD8+ T cells points to a clonal deletion that follows antigen presentation by imiquimod-matured LCs and Langerin+ dDCs lacking the capacity to potently stimulate T cells. Altogether, since Treg are not involved (Idoyaga et al., 2013) and OVA-specific CD8+ T cells are deleted, we do not believe that adoptive transfer experiments would yield significant insights into the mechanism of tolerance induction that we describe in our manuscript.

minor points

1. OVA systems carry the caveat that OTI cells are extremely sensitive to antigen stimulation. Have the authors examined response to other protein/antibody conjugates?

AUTHORS' RESPONSE: We fully agree with the reviewer's caveat. In fact, we do cite a publication (Choi et al., 2009) which emphasizes this point by showing that even picomolar concentrations of OVA can already induce OT-I responses. This is why we chose to avoid the OT-I system for our main readout, i.e. in vivo killing of OVA-loaded target cells by endogenous rather than transgenic OVA-specific CD8+ T cells. We have not, however, tested these responses with other antigen/antibody conjugates. Such conjugates were not available at the time of our study, but were addressed in recently published studies (Idoyaga et al., 2011). Therefore, we feel that this would be beyond the scope of this work.

2. The authors do not indicate the numbers of OTI cells that were transferred within adoptive transfer assays. This is important due to the sensitivity of responses to the numbers of founder cells.

AUTHORS' RESPONSE: We thank the reviewer for this hint and apologize for failing to mention this important information. Injected cell numbers are now indicated in the Materials & Methods section.

3. Could this same system of tolerance induction be applied to demonstrate benefit, e.g. prevention allergy?

AUTHORS' RESPONSE: Allergy responses are controlled by a shift of CD4+ T cells into Th2 cells. We have not monitored whether such Th2 bias could be controlled by immunization with L31/allergen conjugates in the presence of imiquimod. Although this reasoning does make sense, such experiments are beyond the scope of our work, which focused on CD8+ T cell responses.

4. The running title should be reworded: "CD8+ T cells control by Langerin+ skin DCs"

AUTHORS' RESPONSE: We have changed the title accordingly.

5. The statement within the introduction "Once activated, DCs migrate to lymph nodes and prime antigen-specific T cells there. Thus, DCs are essential to initiate adaptive immune responses (Palucka et al., 2010;Steinman, 2012)" does not support the conclusion.

AUTHORS' RESPONSE: This is correct, this statement does not fit with our findings. We have replaced "prime" by "interact with" to avoid implying that priming is an automatic consequence of DC-T interactions in lymph nodes.

6. With the introduction the rather with the text should be removed: "However, since imiquimod triggers emigration and maturation of skin DCs, including LCs (Flacher et al., 2008;Suzuki et al., 2000), its tolerogenic influence appears to rather support alternatives to this paradigm (Joffre et al., 2009)

AUTHORS' RESPONSE: We thank the reviewer for this remark. We have changed the text accordingly.

Referee #3 (Comments on Novelty/Model System):

After the addition of some new data and some better or more thorough explanations, this paper should definitely be published.

Dendritic cells (DC) are a heterologous population of antigen presenting cells that can induce immunity or tolerance depending on the context in which a given antigen is captured. DCs found residing under steady state conditions, or ones that have been activated (matured) by non-inflammatory signals, are generally tolerogenic, although critical unknowns remain regarding what distinguishes DCs that induce tolerance from those that induce immunity. The issue remains a fundamental unknown in our understanding of the immune response and involves problems of both immunology and cell biology, meaning that it is an issue of general interest.

In this manuscript, the authors have extended their previous work on the functional characterization of murine skin DCs. They deliver antigen through the surface lectin langerin to specifically target Langerhans cells (LCs) and dermal DCs in vivo. The data show that dermal DCs are the major DC subset contributing to CD8 T cell priming by Langerin-targeted ovalbumin when injected with the adjuvants polyIC + anti-CD40, whereas LCs do not prime immunity. The data further show that langerin-targeted ovalbumin in combination with a TLR7 agonist (imiquimod) induces tolerance, and that LCs are required for induction of tolerance. This is surprising since imiquimod has previously been shown to induce CD8 T cell immunity and may have important implications as imiquimod is currently used in patients to stimulate immune responses. These results also suggest that TLR7 agonists, and maybe other microbial products, may have either stimulatory or tolerogenic effects depending on the DC subset involved. This is an interesting study that could be strengthened by better characterizing the mechanism underlying these observations.

1. It would be important for the authors to further characterize or at least explain how imiquimod can induce either CD8 T cell immunity or tolerance depending on the context.

Although the authors here show that imiquimod combined with intradermal injection of Langerin-targeted ovalbumin induced tolerance, in a previous report (Flacher et al, 2012), imiquimod combined with intradermal injection of DEC205-

targeted ovalbumin induced CD8 immunity. Langerhans cells were targeted in both vaccination schemes since they express both Langerin and DEC205.

→ It could be that level of antigen presentation by LCs is different following antigen uptake through DEC205 and langerin and determines the capacity of LCs to induce tolerance or immunity.

→ Alternatively, DEC205 could also target other immunostimulatory DC subsets that outcompete LC tolerogenic activity.

AUTHORS' RESPONSE: The points raised by the reviewer address important elements for discussing our results. We have rewritten part of the discussion to incorporate this. We agree that differences in the target receptor result in different internalisation pathways. In addition, these pathways may be different from one DC subset to another.–Considering this, critical variations in antigen processing and presentation can be expected from different targeting strategies.

As suggested by the reviewer, there are indeed several, Langerin-negative DC subsets that can be targeted by DEC-205. We have identified Langerin^{neg} dDCs (Flacher et al., 2012) and CD8+ DCs (Idoyaga et al., 2011) as essential subsets following cutaneous DEC-205 targeting, depending on the mouse strain. This broad and potent population of DCs may well be a key element to explain the differences with Langerin targeting.

In addition, imiquimod may have a greater capacity to directly mature DEC-205+ subsets as compared to Langerin+ subsets, and direct rather than bystander activation has been suggested to potentiate antigen presentation (Blander and Medzhitov, 2006)..

2. Another interesting question not addressed in this study is how LCs and dermal DCs are matured since neither express TLR7 and TLR3 and are thus not directly matured by imiquimod and polyI:C. The authors do refer to this issue in the text, but additional information or discussion is needed to explain or rationalize the mechanism involved.

AUTHORS' RESPONSE: Thus far, no convincing evidence exist that non-immune murine skin cells (fibroblasts, keratinocytes) express TLR7 and provide bystander activation signals to DCs (Drobits et al., 2012). However, mast cells that reside in the dermis represent early responders for TLR7 stimulation (Drobits et al., 2012;Heib et al., 2007), and they play a critical role in attracting pDCs and promoting LC emigration into LNs. Therefore, factors released by mast cells may be critical bystander stimuli that promote migration of skin DCs but lead to suboptimal maturation.

On the other hand, poly(I:C) can be sensed by keratinocytes and fibroblasts, which express TLR3 (Drobits et al., 2012) and possibly cytoplasmic dsRNA receptors, resulting in the release of pro-inflammatory cytokines. This influence of poly(I:C) is probably potentiated by concomitant triggering of CD40, providing a direct maturation signal to DCs. We have extended our discussion with these arguments in the manuscript.

3. Additionally, it has been previously shown that steady-state DCs induce tolerance. Yet here, langerin-targeted OVA alone induces T cell proliferation and exhibits a capacity to induce memory T cells similar to Langerin-targeted OVA combined imiquimod (Figure 1B).

AUTHORS' RESPONSE: Langerin targeting does indeed promote initial proliferation of OT-I transgenic CD8+ T cells, in both untreated and imiquimod-treated mice (Figure 1B). However, dividing T cells of untreated and imiquimod-treated mice show identically low levels of CD127 (Figures 1C and 1D). This lack of IL-7 Receptor expression in both untreated or imiquimod conditions is in line with the poor survival of OVA-specific T cells (Figure 2A). Therefore, it should be considered that the capacity for priming CD8+ T cell responses is poor in both conditions. Conversely, the use of a poly(I:C)/αCD40 combination triggers upregulation of CD127 (Figures 1C and 1D) and long-term survival of OT-I CD8+ T cells (Figure 2A).

Interestingly, despite the similarities in CD8+ T cell responses in untreated and imiquimod-treated mice, tolerance induction is not seen in the absence of an adjuvant (**Figure 5C**). We hypothesize that LCs require imiquimod-induced migration towards LN to elicit the deletion of OVA-specific naïve CD8 T cells. We have incorporated these thoughts into the discussion.

4. In Figure 4A the authors found that LCs did not contribute to CD8 T cell immunity upon immunization with langerin-targeted OVA combined polyIC and anti-CD40. However, they did not explore whether they could also be tolerogenic. This could be addressed by priming LKO→WT mice with langerin-targeted OVA combined with polyIC followed by a boost with DEC-OVA combined with polyIC.

*AUTHORS' RESPONSE: The tolerogenic effects of LCs in the presence of imiquimod are addressed in **Figure 6B** of the original manuscript. There, we pre-immunized LKO→WT chimeric mouse with the langerin-OVA conjugate plus imiquimod and subsequently boosted with DEC-OVA combined with polyIC. In these mice where only LCs could be targeted, we observed a clear tolerance induction. Conversely, when only Langerin+ dermal DCs can be targeted (WT→LKO mice), the tolerogenic effect does not show up.*

*To complement this and fully address the reviewer's concern, we have performed a similar experiment where we pre-immunized in the presence poly(I:C) and anti-CD40. The results of these killing assays are now shown in **Figure E5**. Please note that, due to time constraints, we could not exactly match the conditions described in Figure 6, meaning that the delay between primary and secondary immunization was one week instead of six, but this should not have any critical impact on the outcome. As expected, tolerance induction was absent in control wt mice and in mice where only Langerin+ dDCs could be targeted. On the other hand, chimeric mice with LC-restricted Langerin expression retained a strong tolerogenic induction by targeting of OVA through Langerin. We hypothesize that Langerin-mediated antigen uptake by LCs exclusively and subsequent presentation to CD8+ T cells in skin-draining LNs may be sufficient to promote deletional cross-tolerance in the presence of either adjuvant. However, in wild-type mice, pIC/40 treatment, as opposed to imiquimod, unfolds the strong immunogenic potential of Langerin+ dDCs, thereby hiding the tolerogenic contribution of LCs. In addition, we cannot exclude that the lack of Langerin+ dDCs targeting in LKO→wt mice implies an impaired hypothetical cross-talk between LCs and Langerin+ dDCs in the skin or at the level of lymph nodes, similar to what was described for LCs and LN-resident DCs (Allan et al., 2006).*

5. Figure 4A shows that LCs targeted alone in presence of PolyIC and anti-CD40 did not induce cytotoxic T cells. Nevertheless, the authors were still able to induce proliferation of memory IFN γ + CD8 T cells to the same degree as WT mice. The authors should comment on these results, which appear inconsistent.

*AUTHORS' RESPONSE: We thank the reviewer for this opportunity to clarify our conclusions. First, as also apparent in another experimental setup (**Figure 1B vs. 2A/C**), similar proliferation observed in **Figure 4B** does not predict killing capacity and survival.*

Interestingly, the wt→LKO chimera, in which only Langerin+ dDCs can be targeted, systematically showed a higher capacity of IFN γ production for memory T cells restimulated with OVA peptide 3 weeks after immunization. This is not only consistent with a dominant cross presentation by Langerin+ dDC, but also with a probable lack of dampening of T cell

priming by LCs (**Figure 4D**). Conversely, the reverse chimeras (LKO →wt) showed that significantly less IFN- γ -producing T cells differentiate when only LCs are targeted.

Despite this, as pointed out by the reviewer, we were surprised to observe a similarly low capacity of IFN γ production by memory T cells in wt mice and LKO →wt chimeras. Although IFN-g production may not perfectly correlate with cytotoxicity, it contrasted with the results of killing assays (**Figure 4A**). It should however be noted that this unexpectedly low response of wild-type mice is at odds with previous restimulation assays performed in wt mice after 8 weeks, which yielded ~60% of IFN γ producers among memory OT-I T cells (**see Figure 2C**). Altogether, we believe that this does not change the overall conclusions of our study and the validity of the side-by-side comparison of the chimeric mice.

Minor point:

In the introduction the authors mention that expression of langerin is heterogeneous between mouse strains (Flacher, 2008). It is unclear which DC subsets express Langerin in the mouse strain used in this study and should be clarified.

AUTHORS' RESPONSE: Mice with C57BL/6 genetic background have been used throughout the study. This strain harbours exclusively Langerin expression in the skin DC subsets Langerin+ dDCs and Langerhans cells, but not in the LN-resident CD8+ DCs. We have modified the introduction to state this more clearly.

References

Allan RS, Waithman J, Bedoui S, Jones CM, Villadangos JA, Zhan Y, Lew AM, Shortman K, Heath WR, and Carbone FR (2006) Migratory Dendritic Cells Transfer Antigen to a Lymph Node-Resident Dendritic Cell Population for Efficient CTL Priming. *Immunity* 25: 153-162.

Blander JM and Medzhitov R (2006) Toll-dependent selection of microbial antigens for presentation by dendritic cells. *Nature* 440: 808-812.

Bonifaz LC, Bonnyay DP, Charalambous A, Darguste DI, Fujii SI, Soares H, Brimnes MK, Moltedo B, Moran TM, and Steinman RM (2004) In vivo targeting of antigens to maturing dendritic cells via the DEC-205 receptor improves T cell vaccination. *J Exp Med* 199: 815-824.

Caminschi I, Maraskovsky E, and Heath WR (2012) Targeting Dendritic Cells in vivo for Cancer Therapy. *Front Immunol* 3: 13.

Choi JH, Do Y, Cheong C, Koh H, Boscardin SB, Oh YS, Bozzacco L, Trumpfheller C, Park CG, and Steinman RM (2009) Identification of antigen-presenting dendritic cells in mouse aorta and cardiac valves. *J Exp Med* 206: 497-505.

Clynes RA, Towers TL, Presta LG, and Ravetch JV (2000) Inhibitory Fc receptors modulate in vivo cytotoxicity against tumor targets. *Nat Med* 6: 443-446.

Drobits B, Holcmann M, Amberg N, Swiecki M, Grundtner R, Hammer M, Colonna M, and Sibilio M (2012) Imiquimod clears tumors in mice independent of adaptive immunity by converting pDCs into tumor-killing effector cells. *J Clin Invest* 122: 575-585.

Flacher V, Douillard P, Ait-Yahia S, Stoitzner P, Clair-Moninot V, Romani N, and Saeland S (2008) Expression of langerin/CD207 reveals dendritic cell heterogeneity between inbred mouse strains. *Immunology* 123: 339-347.

Flacher V, Tripp CH, Haid B, Kissenpfennig A, Malissen B, Stoitzner P, Idoyaga J, and Romani N (2012) Skin langerin+ dendritic cells transport intradermally injected anti-DEC-205 antibodies but are not essential for subsequent cytotoxic CD8+ T cell responses. *J Immunol* 188: 2146-2155.

Hawiger D, Inaba K, Dorsett Y, Guo M, Mahnke K, Rivera M, Ravetch JV, Steinman RM, and Nussenzweig MC (2001) Dendritic cells induce peripheral T cell unresponsiveness under steady state conditions in vivo. *J Exp Med* 194: 769-779.

Heib V, Becker M, Warger T, Rechtsteiner G, Tertilt C, Klein M, Bopp T, Taube C, Schild H, Schmitt E, and Stassen M (2007) Mast cells are crucial for early inflammation, migration of Langerhans cells and CTL responses following topical application of TLR7 ligand in mice. *Blood* 110: 946-953.

Idoyaga J, Cheong C, Suda K, Suda N, Kim JY, Lee H, Park CG, and Steinman RM (2008) Cutting Edge: Langerin/CD207 Receptor on Dendritic Cells Mediates Efficient Antigen Presentation on MHC I and II Products In Vivo. *J Immunol* 180: 3647-3650.

Idoyaga J, Fiorese C, Zbytnuik L, Lubkin A, Miller J, Malissen B, Mucida D, Merad M, and Steinman RM (2013) Specialized role of migratory dendritic cells in peripheral tolerance induction. *J Clin Invest* 123: 844-854.

Idoyaga J, Lubkin A, Fiorese C, Lahoud MH, Caminschi I, Huang Y, Rodriguez A, Clausen BE, Park CG, Trumpfheller C, and Steinman RM (2011) Comparable T helper 1 (Th1) and CD8 T-cell immunity by targeting HIV gag p24 to CD8 dendritic cells within antibodies to Langerin, DEC205, and Clec9A. *Proc Natl Acad Sci U S A* 108: 2384-2389.

Idoyaga J, Suda N, Suda K, Park CG, and Steinman RM (2009) Antibody to Langerin/CD207 localizes large numbers of CD8alpha+ dendritic cells to the marginal zone of mouse spleen. *Proc Natl Acad Sci U S A* 106: 1524-1529.

Kreutz M, Tacke PJ, and Figdor CG (2013) Targeting dendritic cells--why bother? *Blood* 121: 2836-2844.

Steinman RM (2012) Decisions about dendritic cells: past, present, and future. *Annu Rev Immunol* 30: 1-22.

Trumpfheller C, Longhi MP, Caskey M, Idoyaga J, Bozzacco L, Keler T, Schlesinger SJ, and Steinman RM (2011) Dendritic cell-targeted protein vaccines: a novel approach to induce T cell immunity. *J Intern Med*.

Thank you for the submission of your revised manuscript to EMBO Molecular Medicine and please accept my sincere apologies for the delay. We have now received the enclosed report from the referee who was asked to re-assess it. As you will see the reviewer is now supportive and I am pleased to inform you that we will be able to accept your manuscript pending final editorial amendments.

Please submit your revised manuscript within two weeks.

I look forward to reading a new revised version of your manuscript as soon as possible.

***** Reviewer's comments *****

Referee #1 (Remarks):

I find the responses satisfactory.